# Nearly Optimal Bounds for Orthogonal Trace-Sum Maximization

## Abstract

Orthogonal trace-sum maximization (OTSM) problems arise in a wide range of data processing applications, including canonical correlation analysis and cryogenic electron microscopy. Despite their practical importance, the existing theoretical understanding of these problems remains incomplete. In this paper, we show that the generalized power method (GPM) converges *linearly* to the global optimal solution with high probability under an additive Gaussian noise model, provided that the noise level satisfies the nearly optimal bound $O(\sqrt{n/\log n})$. In addition, we prove that the semidefinite programming (SDP) relaxation of OTSM is tight and admits a unique optimal solution under the same noise regime, improving upon the best known theoretical guarantee of $O(n^{1/4})$. Extensive numerical experiments further demonstrate that our theoretical predictions closely match empirical observations.

## 1. Introduction

Orthogonal trace–sum maximization (OTSM) problems constitute an important class of data processing problems with widespread applications across science and engineering, including canonical correlation analysis (CCA) (Hanafi & Kiers, 2006; Hotelling, 1992), Procrustes analysis (Gonçalves et al., 2023; Igual et al., 2014; Gower, 2015), and cryogenic electron microscopy (cryo-EM) (Singer & Shkolnisky, 2011; Shkolnisky & Singer, 2012), among many others. A prototypical formulation of OTSM problems takes the form

$$\max \quad \sum_{(i,j)\in\Omega}^{n} \operatorname{tr}(O_i^\top C_{ij} O_j) \qquad \text{(OTSM)}$$
$$\text{s.t.} \quad O_i \in O_{d_i,r}, \ i \in [n]$$

[1]Anonymous Institution, Anonymous City, Anonymous Region, Anonymous Country. Correspondence to: Anonymous Author <anon.email@domain.com>.

Preliminary work. Under review by the International Conference on Machine Learning (ICML). Do not distribute.

where $\Omega \subseteq [n] \times [n]$ denotes the observation set, $O_{d_i,r} := \{O \in \mathbb{R}^{d_i \times r} \mid O^\top O = I_r\}$ (with $d_i \geq r$ and $I_r$ being the $r \times r$ identity matrix) denotes the Stiefel manifold, and the matrices $C_{ij} = C_{ji}^\top \in \mathbb{R}^{d_i \times d_j}$ encode pairwise information between variables. This problem is also commonly referred to as the Stiefel manifold synchronization problem.

The algorithmic approaches to problem (OTSM) primarily fall into two categories: semidefinite relaxation (SDR) methods and first-order iterative methods, most notably the generalized power method (GPM). As shown in (Won et al., 2022), the SDR of (OTSM) can be formulated as

$$\max_{U \succeq 0} \quad \langle C, U \rangle \tag{1}$$
$$\text{s.t.} \quad U_{ii} \preceq I_{d_i}, \ \operatorname{tr}(U_{ii}) = r, \ \forall i \in [n],$$

which lifts the original nonconvex problem into a higher-dimensional convex program by introducing a positive semidefinite matrix variable $U$ and ignoring the rank constraint $\operatorname{rank}(U) \leq r$.

In contrast, the generalized power method (GPM) directly operates on the original nonconvex feasible set and scales favorably to large problem instances. In its simplest form, the GPM iteration is given by

$$G^{t+1} = \pi^n(CG^t) \quad \text{for } t = 0, 1, \ldots,$$

where $G^t = (G_1^t; \ldots; G_n^t)$, and $\pi^n : \mathbb{R}^{(\sum_i d_i) \times r} \to O_{d_1,r} \times \cdots \times O_{d_n,r}$ denotes the blockwise projection onto the product of Stiefel manifolds. Each projection step amounts to computing the leading $r$ singular vectors of the corresponding block, making GPM computationally efficient and easy to implement.

A central question in the study of problem (OTSM) concerns

> when the SDR is tight, and under what conditions the GPM converges to a global optimum, as well as at what rate.

Addressing these questions requires imposing a generative model on the observed matrices $\{C_{ij}\}_{(i,j)\in\Omega}$. A commonly adopted assumption in the literature posits the existence of a collection of ground-truth signals $\{G_i^\star\}_{i=1}^n$ together with noise matrices $\{W_{ij}\}_{(i,j)\in\Omega}$, such that

$$C_{ij} = G_i^\star G_j^{\star\top} + \sigma W_{ij}, \qquad (i,j) \in \Omega, \tag{2}$$

where each $W_{ij} = W_{ji}^\top$ is a Gaussian noise matrix, and $\sigma \geq 0$ controls the noise level. Under this model, the goal is to recover the ground-truth signals $\{G_i^\star\}_{i=1}^n$ from noisy pairwise observations.

Intuitively, one expects that when the observation graph $\Omega$ is sufficiently informative and the noise level $\sigma$ is not too large, near-exact recovery of the ground-truth solution is possible via either SDR or GPM. Characterizing the precise regimes—both in terms of noise magnitude and sampling density—under which such recovery guarantees hold is a central focus of this line of research.

Despite the widespread adoption of the generative model (OTSM), the existing theoretical guarantees remain limited. For the SDR (1), the strongest known tightness results, established in (Won et al., 2022), require the noise level to satisfy $\sigma = O(n^{1/4})$ in the fully observed setting, i.e., when $p = 1$. In contrast, theoretical results for the generalized power method are even more limited. Existing analyses (Won et al., 2021) establish only global convergence to a stationary point, without providing convergence rates or recovery guarantees for the ground-truth solution.

As a result, a substantial gap persists between theory and practice. Empirical evidence suggests that both the semidefinite relaxation and GPM remain effective at significantly higher noise levels, with successful recovery observed when $\sigma = O(\sqrt{n})$ (see the numerical experiments in Section 4). This discrepancy highlights the need for sharper theoretical analyses capable of explaining the strong empirical performance of these methods.

### 1.1. Main Contributions

In this paper, we nearly close the gap between existing theoretical guarantees and empirical observations by establishing linear convergence of the GPM and tightness of the SDR under substantially weaker noise conditions $\sigma = O(\sqrt{n/\log n})$. Our analysis is built upon a careful application of the leave-one-out technique (Zhong & Boumal, 2018; Abbe et al., 2020; Ling, 2022a), which allows us to control delicate dependencies arising from nonconvexity and randomness. Our main contributions can be summarized as follows.

C.1 **Linear convergence of GPM.** We show that, in the fully observed case $p = 1$, the GPM with spectral initialization converges linearly to the global optimal solution of (OTSM) with high probability, provided that $\sigma = O(\sqrt{n/\log n})$.

C.2 **Nearly optimal bounds.** We significantly improve existing guarantees for the SDR by proving tightness in the nearly optimal noise regime $\sigma = O\left(\sqrt{n/\log n}\right)$, improving upon the previous best bound $\sigma = O(n^{1/4})$.

C.3 **Random observation model.** We further extend all of the above results to the random observation setting, where each pairwise measurement is observed independently with probability $p \gtrsim \log n/\sqrt{n}$. In this case, we show that both linear convergence of GPM and tightness of SDR hold when $\sigma = O\left(\sqrt{np/\log n}\right)$.

### 1.2. Related Work

**Synchronization.** The complex-valued synchronization problem corresponding to (OTSM) with $d_1 = \cdots = d_n = r = 1$ is commonly referred to as *angular synchronization* or *phase synchronization* (Bandeira et al., 2017; Boumal, 2016; Liu et al., 2017; Zhong & Boumal, 2018). For this setting, both tightness of the semidefinite relaxation and linear convergence of the generalized power method to the global optimum were established in (Zhong & Boumal, 2018) under the nearly optimal noise regime $\sigma = O(\sqrt{n/\log n})$. The real-valued counterpart with $d_1 = \cdots = d_n = r \geq 2$ is known as the *orthogonal group synchronization* problem (Bandeira et al., 2015; Liu et al., 2023b; Ling, 2022b). In this case, (Ling, 2022b) proved linear convergence of GPM and tightness of the SDP relaxation under the same noise condition $\sigma = O(\sqrt{n/\log n})$.

**Leave-One-Out Technique.** The leave-one-out (LOO) technique is a validation and sensitivity analysis method widely used in statistics, machine learning, and optimization to assess the dependence of an estimator or algorithm on individual data samples. It was first introduced in (Zhong & Boumal, 2018) to analyze the convergence behavior of the generalized power method for phase synchronization, leading to nearly optimal theoretical guarantees. Since then, the LOO technique has been successfully applied to a variety of problems to establish sharp statistical and algorithmic bounds, including entrywise eigenvector analysis (Abbe et al., 2020), top-$K$ ranking (Chen et al., 2019), stochastic block models (Deng et al., 2021), orthogonal group synchronization (Ling, 2022a), and robust orthogonal synchronization (Liu et al., 2023a).

**Notation.** For notational simplicity, we define $d := \frac{1}{n}\sum_{i=1}^n d_i$, which is assumed to be independent of $n$. Define

$$O_{d,r}^n = O_{d_1,r} \times \cdots \times O_{d_n,r}.$$

We use $O(d) = O_{d,d}$ to denote the set of orthogonal matrices with dimension $d$. We represent the collection of variables $G_i \in O_{d_i,r}$ by stacking them into a single matrix

$$G = (G_1; \cdots; G_n) \in \mathbb{R}^{nd\times r}.$$

For any matrix $X \in \mathbb{R}^{nd\times nd}$, we denote by $X_{ij} \in \mathbb{R}^{d_i\times d_j}$ its $(i, j)$-th block. Throughout the paper, the notation $x \gtrsim y$ and $x \lesssim y$ means that there exists a constant $c > 0$,

independent of $n$ and $p$, such that $x \geq c\,y$ and $x \leq c\,y$, respectively. Besides, we shall use $\mathrm{BlkDiag}(\boldsymbol{X}_1, \ldots, \boldsymbol{X}_K)$ to denote the block diagonal matrix whose diagonal blocks are $\boldsymbol{X}_1, \ldots, \boldsymbol{X}_K$.

## 2. Preliminaries

### 2.1. Background

In this section, we use canonical correlation analysis (CCA) as a representative example to illustrate the background and practical relevance of orthogonal trace–sum maximization problems. Additional examples and applications can be found in (Won et al., 2021).

Canonical correlation analysis (Hotelling, 1992) is a classical multivariate statistical technique that seeks linear projections maximizing the correlation between two sets of observed variables. Given two data matrices $A_1 \in \mathbb{R}^{n \times d_1}$ and $A_2 \in \mathbb{R}^{n \times d_2}$, CCA solves

$$\max_{t_1, t_2} \quad \mathrm{corr}(A_1 t_1, A_2 t_2)$$
$$\text{s.t.} \quad \|t_i\|_2 = 1, \quad i = 1, 2, \tag{3}$$

where $t_i \in \mathbb{R}^{d_i}$ are the projection vectors, and $\mathrm{corr}(\cdot, \cdot)$ denotes the Pearson correlation coefficient.

Several important generalizations of CCA have been proposed. First, the framework can be extended to handle more than two datasets $A_1, \ldots, A_n$ with $n \geq 2$. Second, instead of seeking single projection vectors, one may optimize over *partial orthogonal transformations* to align the datasets in a common low-dimensional subspace. Prominent examples include the *MAXDIFF* and *MAXBET* criteria (Gonçalves et al., 2023; Igual et al., 2014; Gower, 2015), which solve the following orthogonality-constrained trace optimization problems:

$$\max_{\{O_i\}} \quad \sum_{i<j} \mathrm{tr}\big(O_i^\top A_i^\top A_j O_j\big)$$
$$\text{s.t.} \quad O_i \in O_{d_i, r}, \quad i = 1, \ldots, n, \tag{MAXDIFF}$$

and

$$\max_{\{O_i\}} \quad \frac{1}{2} \sum_{i,j=1}^{n} \mathrm{tr}\big(O_i^\top A_i^\top A_j O_j\big)$$
$$\text{s.t.} \quad O_i \in O_{d_i, r}, \quad i = 1, \ldots, n. \tag{MAXBET}$$

Both MAXDIFF and MAXBET can be viewed as special instances of (OTSM) by setting $S_{ij} = A_i^\top A_j$ (with $S_{ii} = 0$ for MAXDIFF).

### 2.2. Algorithm

In this section, we review the spectral initialization procedure and the generalized power method (GPM) for solving problem (OTSM).

---

**Algorithm 1** GPM for stiefel manifold synchronization

---

**Input:** Measurements matrix $C \in \mathbb{R}^{nd \times nd}$, size $r$
**Initialization:** Compute the top $r$ eigenvectors $\widetilde{G} \in \mathbb{R}^{nd \times r}$ of $C$ and set $G^0 = \pi^n(\widetilde{G})$
**for** $t = 0, 1, \cdots$ **do**
    Set $G^{t+1} = \pi^n(CG^t)$
    Stop if $\|G^{t+1} - G^t\|_F$ is small enough
**end for**

---

**Spectral initialization.** Spectral initialization is a relaxation-based technique that constructs an initial point by exploiting the spectral structure of the information matrix $C$. Specifically, we compute the leading $r$ eigenvectors of $C$, denoted by $\widetilde{G} \in \mathbb{R}^{nd \times r}$, and then project $\widetilde{G}$ onto the product of Stiefel manifolds. Let

$$\pi^n : \mathbb{R}^{nd \times r} \to O_{d_1, r} \times \cdots \times O_{d_n, r}$$

denote the blockwise projection operator. For each block $i$, the projection $\pi_i : \mathbb{R}^{d_i \times r} \to O_{d_i, r}$ is given by

$$\pi_i(\widetilde{G}_i) = P_i Q_i^\top,$$

where $P_i \in \mathbb{R}^{d_i \times r}$ and $Q_i \in \mathbb{R}^{r \times r}$ are the left and right singular vectors of $\widetilde{G}_i$, respectively.

**Generalized power method.** The generalized power method was introduced in (Journee et al., 2008) as an iterative procedure for maximizing a convex function over a compact feasible set. At each iteration, GPM maximizes an affine minorant of the objective function at the current iterate over the feasible set to produce the next iterate. In the context of problem (OTSM), the standard GPM update takes the form

$$G^{t+1} = \pi^n\left(\left(I + \frac{\alpha}{n}C\right)G^t\right),$$

where $\alpha \in (0, \infty]$ is a stepsize parameter. In Algorithm 1, we focus on the special case $\alpha = \infty$, which reduces to a power-iteration-type update analogous to the classical power method in the vector setting.

### 2.3. Problem Setting

We assume that the observation set $\Omega$ follows the well-known Erdős–Rényi model $\mathcal{G}(n, p)$, which means that each $(i, j) \in [n] \times [n]$ is observed with probability $p \in (0, 1]$, independently from every other pair.

Since the measurements $\{C_{ij} : (i, j) \in \Omega\}$ in (OTSM) are invariant under multiplication of a comment rotation in $O(r)$ to the target signal $G^\star$, we can only identify $G^\star$ up to a global rotation. This motivates us to define the following quotient space.

**Definition 2.1** (Quotient space). For two matrices $A, B \in \mathbb{R}^{nd \times r}$, we define equivalent relationship $\sim$ by

$$A \sim B \iff \exists Q \in O(r), \ st. \ A = BQ.$$

In the quotient space $\mathbb{R}^{nd \times r} / \sim$, we define metric $d_F$ by

$$d_F([A], [B]) = d_F(A, B) = \min_{Q \in O(r)} \|A - BQ\|_F.$$

We claim that the sub space $(O_{d,r}^n / \sim, d_F)$ is a complete metric space.

In our analysis, we also need the following concept of block-wise norm and block-wise distance.

**Definition 2.2** (Block-wise norm and distance). The block-wise norm $\|\cdot\|_\infty$ on $\mathbb{R}^{nd \times r}$ is defined as

$$\|A\|_\infty = \max_{i \in [n]} \|A_i\|_F, \quad \forall A \in \mathbb{R}^{nd \times r}$$

where $A_i \in \mathbb{R}^{d_i \times r}$ is the $i$-th block of $A$. The block-wise distance $d_\infty$ is defined as

$$d_\infty(A, B) = \min_{Q \in O(r)} \|A - BQ\|_\infty, \quad \forall A, B \in \mathbb{R}^{nd \times r}.$$

## 3. Main Results

In this section, we present our main theoretical results. The following theorem summarizes our findings, establishing that the generalized power method converges at a linear rate to the unique (up to a global rotation) global optimal solution $G^\infty$ to the problem (OTSM) with high probability. Moreover, we show that the semidefinite relaxation (1) is tight and that $G^\infty (G^\infty)^\top$ is the unique optimal solution of the SDP problem (1).

**Theorem 3.1** (Overall). *Suppose that the sampling ratio $p$ and the noise level $\sigma$ satisfy*

$$p \gtrsim \frac{\log n}{\sqrt{n}} \quad and \quad \sigma \lesssim \sqrt{\frac{np}{\log n}}. \tag{4}$$

*Then, with probability at least $1 - O(n^{-2})$, the following statements hold:*

*(1) The GPM converges linearly to the global optimal solution $G^\infty$ of (OTSM), in the sense that*

$$d_F\left(G^{t+1}, G^\infty\right) \leq \frac{1}{2} d_F\left(G^t, G^\infty\right), \qquad \forall t \geq 0.$$

*(2) The global optimal solution $G^\infty$ satisfies*

$$d_F(G^\infty, G^\star) \lesssim \frac{1+\sigma}{\sqrt{p}}, \ d_\infty(G^\infty, G^\star) \lesssim (1+\sigma)\sqrt{\frac{\log n}{np}}.$$

*(3) The semidefinite relaxation (1) is tight, and $G^\infty (G^\infty)^\top$ is the unique optimal solution to (1).*

*Remark* 3.2. The condition $\sigma \lesssim \sqrt{np/\log n}$ is nearly optimal up to logarithmic factors and matches empirical observations reported in the literature. In particular, when $p = 1$, the theorem provides recovery guarantees under noise levels $\sigma = O(\sqrt{n/\log n})$, significantly improving upon the previously best bound $\sigma = O(n^{1/4})$ (Won et al., 2022).

- Part (1) establishes a global linear convergence rate for GPM, which is nontrivial given the nonconvex nature of the OTSM problem. The result shows that, when initialized spectrally, GPM not only avoids spurious stationary points but also contracts geometrically toward the global optimum $G^\infty$.

- Part (2) quantifies the statistical accuracy of the recovered solution relative to the ground truth $G^\star$. Moreover, in the fully observed setting ($p = 1$), the bounds in both the Frobenius norm and the blockwise infinity norm match the optimal rates established for phase synchronization (Zhong & Boumal, 2018) and orthogonal group synchronization (Ling, 2022b). This agreement strongly suggests that the bounds in Part (2) are nearly information-theoretically optimal.

- Part (3) demonstrates that the semidefinite relaxation is tight in the same noise regime and admits a unique optimal solution. This result establishes a strong connection between the global optimum of the nonconvex formulation and the convex SDP relaxation, providing a unified explanation for the success of both approaches.

Taken together, the theorem provides a unified theoretical framework that simultaneously explains the convergence behavior of GPM and the tightness of SDP relaxations under nearly optimal conditions, thereby substantially narrowing the gap between theory and practice for orthogonal trace-sum maximization problems.

### 3.1. Full Observation Model

In this section, we focus on the fully observed setting, namely $p = 1$, where the observation set satisfies $\Omega = [n] \times [n]$, so we have

$$C = G^\star G^{\star\top} + \sigma W. \tag{5}$$

All results will be extended to the randomly observed setting in the subsequent section.

#### 3.1.1. LEAVE-ONE-OUT TECHNIQUE

In this section, we explain how the leave-one-out (LOO) technique is employed in our analysis. As will become clear, the main technical challenge lies in deriving a sharp bound for the quantity $\|WG^t\|_\infty$.

*Core difficulty.* Due to the mechanisms of spectral initialization and the GPM updates, all iterates $\{G^t\}_{t \geq 0}$ are intrinsically dependent on the noise matrix $W$.

This dependence renders many standard concentration inequalities inapplicable, as they typically rely on independence between random variables. A naive bound of the quantity $\|WG^t\|_\infty$ obtained via

$$\|WG^t\|_\infty = \max_{i \in [n]} \|w_i^\top G^t\|_F \leq \max_{i \in [n]} \|w_i\|_2 \|G^t\|_F \lesssim n \tag{6}$$

is too crude for our purposes. Here, $w_i \in \mathbb{R}^{nd \times d_i}$ denotes the $i$-th block column of $W$, and the last inequality follows from the high-probability bound $\|w_i\|_2 \lesssim \sqrt{n}$ together with $\|G^t\|_F = \Theta(\sqrt{n})$.

To overcome this difficulty, we adopt the leave-one-out technique by introducing the following auxiliary objects:

- $W^{(m)} \in \mathbb{R}^{nd \times nd}$, obtained from $W$ by setting the $m$-th block row and block column to zero;

- $C^{(m)} = G^\star G^{\star\top} + \sigma W^{(m)}, \quad \Delta W^{(m)} = W - W^{(m)}$;

- $\{G^{t,m}\}_{t \geq 0}$, the sequence generated by the GPM with $C$ replaced by $C^{(m)}$, and $\widetilde{G}^{(m)}$, the matrix formed by the leading $r$ eigenvectors of $C^{(m)}$, normalized so that $(\widetilde{G}^{(m)})^\top \widetilde{G}^{(m)} = nI_r$.

Let $d_F(G^t, G^{t,m}) = \min_{Q \in O(r)} \|G^t Q - G^{t,m}\|_F$. By the triangle inequality, we obtain

$$\|w_m^\top G^t\|_F \leq \|w_m^\top G^{t,m}\|_F + \|w_m^\top (G^t Q - G^{t,m})\|_F$$
$$\leq \|\Delta W^{(m)} G^{t,m}\|_F + \|w_m\|_2 \, d_F(G^t, G^{t,m}),$$

where the second inequality uses the fact that $w_m^\top G^{t,m}$ is a block of $\Delta W^{(m)} G^{t,m}$.

Crucially, $\Delta W^{(m)}$ is independent of $G^{t,m}$, which allows us to apply concentration inequalities to show that $\|\Delta W^{(m)} G^{t,m}\|_F = O(\sqrt{n \log n})$ with high probability (see Lemma 3.4). Moreover, since $C$ and $C^{(m)}$ differ only in a single block row and column, we can show that $d_F(G^t, G^{t,m}) = O(1)$ uniformly in $t$ (see Theorem E.2). Combining these bounds with $\|w_m\|_2 \lesssim \sqrt{n}$, we conclude that

$$\|WG^t\|_\infty \lesssim \sqrt{n \log n},$$

which is much sharper than the naive bound given by (6), and is a key estimate underpinning our convergence and recovery analysis.

### 3.1.2. SPECTRAL INITIALIZATION

We begin with the following theorem, which guarantees that the spectral initialization produces a nearly optimal estimate.

**Theorem 3.3** (Initialization). *Let $\widetilde{G} \in \mathbb{R}^{nd \times r}$ be the top $r$ leading eigenvectors of $C$, $\|\widetilde{G}\|_F = \sqrt{nr}$. Then there exists absolute constants $c_0, C_0 > 0$, such that when $\sigma \leq c_0 \sqrt{n / \log n}$, with probability at least $1 - O(n^{-2})$, the following statements hold:*

$$d_F(\widetilde{G}, G^\star) \leq C_0 \sigma, \quad d_\infty(\widetilde{G}, G^\star) \leq C_0 \sigma \sqrt{\log n / n}. \tag{7}$$

Theorem 3.3 shows that the spectral initialization lies within a small neighborhood of the ground-truth solution $G^\star$, which is a key prerequisite for establishing fast global convergence of the generalized power method. In fact, the size of the guaranteed neighborhood matches, up to constant factors, that of the global optimum characterized in Theorem 3.1, although the constants are generally conservative. This behavior is consistent with existing results for orthogonal synchronization (Ling, 2022a), where spectral methods are shown to achieve nearly optimal statistical accuracy for synchronization over orthogonal and permutation groups.

To proceed, define $U^{(m)} \subseteq \mathbb{R}^{nd \times r}$, $1 \leq m \leq n$, be collections of random matrices satisfying, for all $u \in U^{(m)}$, $\|u\|_F = \sqrt{nr}$ and $u$ is independent of $\Delta W^{(m)}$. Moreover, assume that for each $m$, the cardinality of $U^{(m)}$ is deterministic and bounded by $3n^2$.

A key ingredient of our analysis is a sharp bound on the quantity $\max_{u \in U^{(m)}} \|\Delta W^{(m)} u\|_F$ where we take either $U^{(m)} = \{\widetilde{G}^{(m)}\}$ or $U^{(m)} = \{G^{t,m} : 0 \leq t \leq 3n^2 - 1\}$, which allows us to obtain an $O(\sqrt{n \log n})$ upper bound based on the following lemma.

**Lemma 3.4.** *With probability at least $1 - O(n^{-2})$, the following holds for all $1 \leq m \leq n$:*

$$\max_{u \in U^{(m)}} \|\Delta W^{(m)} u\|_F \leq C_1 \sqrt{n \log n} + C_1 \sqrt{n} \, M_m, \tag{8}$$

*where $C_1 > 0$ is an absolute constant and $M_m := \max_{u \in U^{(m)}} \|u_m\|_F$, with $u_m \in \mathbb{R}^{d_m \times r}$ denoting the $m$-th block of $u$.*

### 3.1.3. LOCAL CONCENTRATION AND CONVERGENCE

The following lemma constitutes a central step in our analysis. It shows that the iteration mapping

$$\mathcal{P}_C : O_{d,r}^n \to O_{d,r}^n, \qquad \mathcal{P}_C(G) = \pi^n \left( \frac{1}{n} CG \right),$$

is locally contractive in a suitable neighborhood of the ground-truth solution.

**Lemma 3.5** (Local contractivity of $\mathcal{P}_C$). *Let $C = G^\star G^{\star\top} + \sigma W$ and $C' = G^\star G^{\star\top} + \sigma W'$. For $x, y \in O_{d,r}^n$, suppose there exist positive constants $k_1, k_2, \epsilon_1, \epsilon_2$ such that*

$$d_F(x, G^\star) \le \epsilon_1 \sqrt{nr}, \quad \|Wx\|_\infty \le k_1 \sqrt{n \log n},$$
$$d_F(y, G^\star) \le \epsilon_2 \sqrt{nr}, \quad \|W'y\|_\infty \le k_2 \sqrt{n \log n}. \quad (9)$$

*Assume*

$$\epsilon := \min\left\{ \frac{\epsilon_1^2 r}{2} + \sigma k_1 \sqrt{\frac{\log n}{n}}, \ \frac{\epsilon_2^2 r}{2} + \sigma k_2 \sqrt{\frac{\log n}{n}} \right\} < 1.$$

*Then we have*

$$d_F(\mathcal{P}_C(x), \mathcal{P}_{C'}(y)) \le \frac{4}{(1-\epsilon)n} d_F(Cx, C'y). \quad (10)$$

*Moreover, if $C = C'$, then*

$$d_F(\mathcal{P}_C(x), \mathcal{P}_C(y)) \le \rho \, d_F(x, y), \quad (11)$$

*where*

$$\rho := \frac{4}{1-\epsilon} \left( 2 \max\{\epsilon_1, \epsilon_2\} \sqrt{r} + \frac{\sigma \|W\|_2}{n} \right).$$

Using the above local contractivity, we can refine the accuracy of the projected spectral initialization $G^0 = \pi^n(\widetilde{G})$.

**Proposition 3.6.** *Under the conditions of Theorem 3.3, with probability at least $1 - O(n^{-2})$,*

$$d_F(G^0, G^\star) \lesssim \sigma, \quad d_\infty(G^0, G^\star) \lesssim \sigma \sqrt{\frac{\log n}{n}}. \quad (12)$$

The next theorem shows that the sequence generated by GPM remains within a neighborhood of the ground truth where the iteration map $\mathcal{P}_C$ is contractive. We refer to this neighborhood as the *contractive region*.

**Theorem 3.7** (Contractive region). *Assume $\sigma \lesssim \sqrt{n/\log n}$. Then, with probability at least $1 - O(n^{-2})$, for all $t \in \{0, 1, \ldots, 3n^2 - 1\}$,*

$$d_F(G^t, G^\star) \le k_3 \sqrt{n}, \quad (13)$$
$$d_F(G^{t,m}, G^t) \le k_1, \quad \forall m \in [n], \quad (14)$$
$$\|WG^t\|_\infty \le k_2 \sqrt{n \log n}, \quad (15)$$

*where $k_1, k_2, k_3 > 0$ are absolute constants.*

Since the iterates $G^t$ remain in the contractive region, the mapping $\mathcal{P}_C$ is contractive along the trajectory of the algorithm. As a consequence, $\{G^t\}$ forms a Cauchy sequence in the quotient space $(O_{d,r}^n / \sim, d_F)$. This leads to the following convergence result.

**Theorem 3.8** (Convergence). *Under the conditions of Theorem 3.7, with probability at least $1 - O(n^{-2})$,*

$$d_F(G^{t+1}, G^t) \le \frac{1}{2} d_F(G^t, G^{t-1}), \quad \forall t \ge 0. \quad (16)$$

*Moreover, in the metric space $(O_{d,r}^n / \sim, d_F)$, the equivalence classes $[G^t]$ converge linearly to a limit $[G^\infty]$ at rate $1/2$, namely,*

$$d_F(G^{t+1}, G^\infty) \le \frac{1}{2} d_F(G^t, G^\infty), \quad \forall t \ge 0, \quad (17)$$

*where $G^\infty \in O_{d,r}^n$ is a fixed point of the mapping $\mathcal{P}_C$.*

### 3.1.4. VERIFYING OPTIMALITY

To verify the optimality of $G^\infty$, we invoke a dual certificate for the semidefinite relaxation. The following optimality conditions are adapted from (Won et al., 2021).

**Lemma 3.9.** *Let $O = (O_1; \cdots; O_n) \in O_{d,r}^n$. Suppose there exist symmetric matrices $\Lambda_i \in \mathbb{R}^{r \times r}$, $i \in [n]$, such that $(CO)_i = O_i \Lambda_i$. Let $\tau_i$ denote the smallest eigenvalue of $\Lambda_i$, and define*

$$L(O, \Lambda) = \mathrm{BlkDiag}\Big( O_1 \Lambda_1 O_1^\top + \tau_1(I_{d_1} - O_1 O_1^\top),$$
$$\ldots, O_n \Lambda_n O_n^\top + \tau_n(I_{d_n} - O_n O_n^\top) \Big) - C.$$

*If $L(O, \Lambda)$ is positive semidefinite, then $O$ is a global optimal solution of* (OTSM). *Moreover, if $\mathrm{rank}(L(O, \Lambda)) = nd - r$, then $O$ is the unique global maximizer of* (OTSM), *and $OO^\top$ is the unique global maximizer of the SDP relaxation.*

We are now ready to establish the main optimality result for the fully observed setting.

**Theorem 3.10.** *Under the conditions of Theorem 3.7, with probability at least $1 - O(n^{-2})$, the limit point $G^\infty$ is the unique global maximizer of* (OTSM) *and satisfies*

$$d_F(G^\infty, G^\star) \lesssim \sigma, \quad d_\infty(G^\infty, G^\star) \lesssim \sigma \sqrt{\frac{\log n}{n}}.$$

*Moreover, $G^\infty (G^\infty)^\top$ is the unique global maximizer of the semidefinite relaxation.*

### 3.2. Random Observation Model

In this section, we consider the random observation model, in which each block on and above the diagonal of the information matrix $C$ in the fully observed setting is independently observed with probability $p$.

Let $E = [e_{ij}]_{(i,j) \in [n] \times [n]}$ be a symmetric random matrix, where $e_{ij}$ are Bernoulli random variables satisfying $\mathbb{P}(e_{ij} = 1) = p$ and $\mathbb{P}(e_{ij} = 0) = 1 - p$. The entries of $E$ on

and above the diagonal are mutually independent, and $E$ is independent of the noise matrix $W$. Define

$$(G^\star G^{\star\top})_\star := [\, e_{ij} G_i^\star G_j^{\star\top} \,]_{(i,j)\in[n]\times[n]},$$

$$W_\star := [\, e_{ij} W_{ij} \,]_{(i,j)\in[n]\times[n]}.$$

Since the observation pattern follows an Erdős–Rényi model, we have $\mathbb{E}\big[(G^\star G^{\star\top})_\star\big] = p\, G^\star G^{\star\top}$. Define the fluctuation term

$$\Delta_\star := (G^\star G^{\star\top})_\star - p\, G^\star G^{\star\top}.$$

Then the information matrix can be decomposed as

$$C_\star = p\, G^\star G^{\star\top} + \Delta_\star + \sigma W_\star. \qquad (18)$$

Next, define the rescaled matrix $C_p := C_\star/p$, namely,

$$C_p = G^\star G^{\star\top} + \frac{1}{p}\Delta_\star + \frac{\sigma}{p}W_\star. \qquad (19)$$

This formulation reduces the problem to a fully observed model with additive noise

$$\widetilde{W} := \frac{1}{p}\Delta_\star + \frac{\sigma}{p}W_\star.$$

The noise structure is the primary distinction between the fully observed and randomly observed models. In the analysis of the fully observed case, all required properties of the noise are encapsulated in Lemma A.1 and Lemma 3.4. Since analogous bounds are available in the random observation setting, as established in Lemma H.1 and Lemma 3.12, the arguments developed for the fully observed model extend directly to the random observation model.

**Theorem 3.11.** *Assume that $p \gtrsim \frac{\log n}{\sqrt{n}}$ and $\sigma \lesssim \sqrt{np/\log n}$. Then, with probability at least $1-O(n^{-2})$, the generalized power method converges linearly to the global optimal solution of* (OTSM), *and the semidefinite relaxation is tight. Moreover, the limiting solution $G^\infty$ satisfies*

$$d_F(G^\infty, G^\star) \lesssim \frac{1+\sigma}{\sqrt{p}}, \quad d_\infty(G^\infty, G^\star) \lesssim (1+\sigma)\sqrt{\frac{\log n}{np}}.$$

A key ingredient of our analysis is still a sharp bound on the quantity $\max_{u\in U^{(m)}} \|\Delta \widetilde{W}^{(m)} u\|_F$ where we take either $U^{(m)} = \{\widetilde{G}^{(m)}\}$ or $U^{(m)} = \{G^{t,m} : 0 \le t \le 3n^2 - 1\}$, which allows us to obtain an $O(\sqrt{np\log n})$ upper bound based on the following lemma.

**Lemma 3.12.** *Assume that $\sigma \lesssim \sqrt{np/\log n}$ and $p \gtrsim \frac{\log n}{\sqrt{n}}$. Then, with probability at least $1 - O(n^{-2})$,*

$$\forall\, 1 \le m \le n, \quad \max_{u\in U^{(m)}} \|\Delta W_\star^{(m)} u\|_F \lesssim \sqrt{np\log n},$$

$$\forall\, 1 \le m \le n, \quad \max_{u\in U^{(m)}} \|\Delta (\Delta_\star)^{(m)} u\|_F \lesssim \sqrt{np\log n}.$$

$$(20)$$

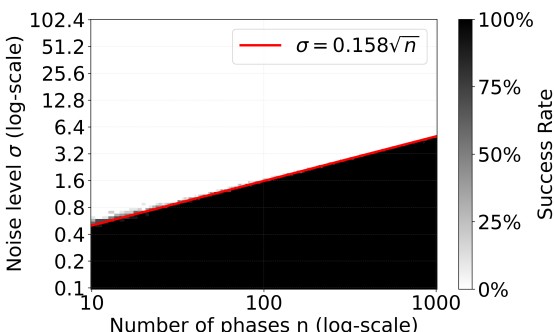

*Figure 1.* Fully observed model: success rate of GPM converging to the global optimum of OTSM.

## 4. Experimental Results

### 4.1. Verifying Convergence of GPM

In this section, we conduct numerical experiments to verify that the generalized power method (GPM) converges to the global optimum of the OTSM problem under the noise regimes predicted by our theory.

**Fully observed model.** We generate 100 values of the noise level $\sigma$ ranging from 0.1 to 100, together with 200 values of $n$ ranging from 10 to 1000. For each pair $(n, \sigma)$, we generate 100 independent OTSM instances with dimensions $d = 10$ and $r = 3$ according to the Gaussian noise model (5). Each instance is solved using the GPM (Algorithm 1). The algorithm is terminated when the Frobenius norm of the difference between two consecutive iterates falls below $10^{-8}$, or when the maximum number of iterations reaches 400.

After obtaining the GPM estimator $\widehat{G}^\infty$, we verify its global optimality using the dual certificate in Lemma 3.9. Specifically, a solution $\widehat{G}^\infty$ is declared optimal if the corresponding matrix $L = L(\widehat{G}^\infty, \Lambda)$ is positive semidefinite. Due to numerical precision, we treat $L$ as positive semidefinite if

$$\frac{\lambda_{\min}(L)}{|\lambda_{\max}(L)|} \ge -10^{-5},$$

which is consistent with the criterion in (Boumal, 2016). For each pair $(n, \sigma)$, the success rate is computed as the fraction of instances (out of 100) for which GPM converges to a certified global optimum. The results are shown in Figure 1.

In Figure 1, the horizontal axis corresponds to $n$ and the vertical axis to $\sigma$, both displayed on logarithmic scales. The shading at each $(n, \sigma)$ indicates the success rate, with black representing a success rate of 1 and white representing 0. The red reference line corresponds to $\sigma = 0.158\sqrt{n}$. The figure shows that GPM successfully converges to the global optimum as long as the noise level remains below approximately $0.158\sqrt{n}$, which is in excellent agreement with our theoretical predictions.

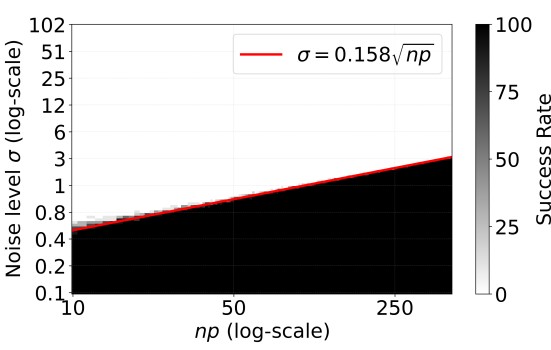

*Figure 2.* Random observation model: success rate of GPM converging to the global optimum of OTSM.

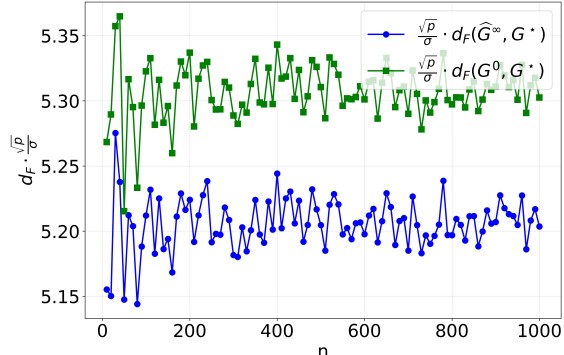

*Figure 3.* Comparison of $\frac{\sqrt{p}}{\sigma}d_F(\widehat{G}^\infty, G^\star)$ and $\frac{\sqrt{p}}{\sigma}d_F(G^0, G^\star)$.

**Random observation model.** We next consider the random observation setting and fix the observation probability to be

$$p = 2\log n/\sqrt{n},$$

and repeat the above experiment using the same ranges of $n$ and $\sigma$. For visualization, we plot the success rate with $np$ on the horizontal axis and $\sigma$ on the vertical axis. The results are shown in Figure 2.

As shown in Figure 2, GPM successfully converges to the global optimum until the noise level reaches approximately $0.158\sqrt{np}$, again closely matching our theoretical guarantees.

## 4.2. Verifying the Bound $d_F(G^\infty, G^\star) \lesssim \frac{1+\sigma}{\sqrt{p}}$

We now empirically verify the error bound established in Theorem 3.11. We generate 100 values of $n$ ranging from 10 to 1000. For each $n$, we generate 10 random observation OTSM instances with parameters $d = 10$, $r = 3$, observation probability and noise level

$$p = 2\log n/\sqrt{n}, \quad \sigma = 0.15\sqrt{np}.$$

We apply GPM to obtain the estimator $\widehat{G}^\infty$ and compute the average recovery error $d_F(\widehat{G}^\infty, G^\star)$ over the 10 instances. For comparison, we also compute the average error of the spectral initialization $d_F(G^0, G^\star)$.

To illustrate the scaling behavior, we plot $n$ on the horizontal axis and $\frac{\sqrt{p}}{\sigma}d_F(\widehat{G}^\infty, G^\star)$ together with $\frac{\sqrt{p}}{\sigma}d_F(G^0, G^\star)$ on the vertical axis. The results are shown in Figure 3.

From Figure 3, we observe that as $n$ varies from 10 to 1000, both $\frac{\sqrt{p}}{\sigma}d_F(\widehat{G}^\infty, G^\star)$ and $\frac{\sqrt{p}}{\sigma}d_F(G^0, G^\star)$ remain essentially constant, thereby validating the predicted $O\left(\frac{1+\sigma}{\sqrt{p}}\right)$ error bound. Moreover, the results clearly indicate that the GPM estimator $\widehat{G}^\infty$ achieves a consistent improvement over the spectral initialization $G^0$.

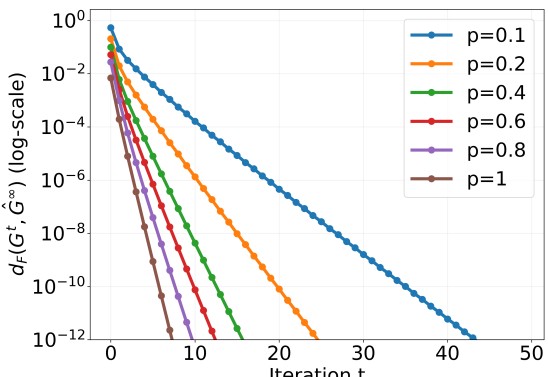

*Figure 4.* Linear convergence of GPM

## 4.3. Verifying Linear Convergence Rate of GPM

To verify the linear convergence of GPM, we generate five OTSM instances with parameters $n = 100$, $d = 10$, $r = 3$, and observation probabilities $p \in \{0.1, 0.2, 0.4, 0.6, 0.8, 1\}$. For each instance, we run GPM for 400 iterations and use the iterate at the final step as an approximation of the limit point, namely $\widehat{G}^\infty := G^{400}$. We then compute the distance $d_F(G^t, \widehat{G}^\infty)$ for the first 50 iterations. The results are shown in Figure 4. As can be observed, $\log d_F(G^t, \widehat{G}^\infty)$ decreases linearly with respect to the iteration index $t$, which provides clear empirical evidence of the linear convergence of the generalized power method.

## 5. Conclusion

This paper establishes nearly optimal theoretical guarantees for orthogonal trace–sum maximization problems. Using the leave-one-out technique, we prove global linear convergence of the generalized power method and tightness of the semidefinite relaxation under sharp noise conditions, for both fully observed and randomly observed settings. Numerical experiments confirm that the theoretical predictions accurately capture empirical performance.

## Impact Statement

This paper presents work whose goal is to advance the field of machine learning. There are many potential societal consequences of our work, none of which we feel must be specifically highlighted here.

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

# Appendices

## A. Proof of Lemma 3.4

Several operator-norm bounds for the noise matrices are crucial in the analysis, summarized in the following lemmas.

**Lemma A.1.** *With probability at least $1 - O(n^{-2})$, the following bounds hold simultaneously:*

$$\|W\|_2 \leq C_2\sqrt{n}, \quad \|W^{(m)}\|_2 \leq C_2\sqrt{n}, \quad \|\Delta W^{(m)}\|_2 \leq C_2\sqrt{n}, \quad \|w_m\|_2 \leq C_2\sqrt{n}, \tag{21}$$

*where $C_2 > 0$ is an absolute constant.*

*Proof.* Using corollary 4.4.8 in (Vershynin, 2018), we have that with probability at least $1 - 4exp(-t^2)$,

$$\|W\|_2 \leq C(\sqrt{nd} + t). \tag{22}$$

Taking $t = \sqrt{nd}$, we get the desired result. For $W^{(m)}$ and $\Delta W^{(m)}$ we can prove the bounds similarly.

For $w_m, \forall x \in \mathbb{R}^{nd}, \|x\|_2 = 1$

$$\|w_m^\top x\|_2 \leq \|Wx\|_2 \leq \|W\|_2, \tag{23}$$

where the first inequality is because all elements of $w_m^\top x$ are a part of vector $Wx$. Therefore $\|w_m\|_2 = \|w_m^\top\|_2 \leq \|W\|_2$. □

*Proof of Lemma 3.4.* We only need to prove that for a fixed $m$ and a non-random $U^{(m)}$, with probability at least $1 - O(n^{-3})$ , $\max_{u \in U^{(m)}} \|\Delta W^{(m)}u\|_F \leq C_1\sqrt{n\log n} + C_1\sqrt{n}M_m$. The reasons are as follows. First, if we can prove the inequality for non-random $U^{(m)}$, then leveraging independence of $\Delta W^{(m)}$ and $U^{(m)}$, we can prove for random $U^{(m)}$ by taking conditional probability and using law of Total Probability. Second, if we can prove the inequality for fixed $m$ with probability at least $1 - O(n^{-3})$, then by taking a union bound over $m \in [n]$, we can derive inequality (8) with probability at least $1 - O(n^{-2})$.

Now, we prove that for a fixed $m$ and a non-random $U^{(m)}$, with probability at least $1 - O(n^{-3})$ , $\max_{u \in U^{(m)}} \|\Delta W^{(m)}u\|_F \leq C_1\sqrt{n\log n} + C_1\sqrt{n}M_m$.

let $u \in U^{(m)}$, we have:

$$\|\Delta W^{(m)}u\|_F^2 = \|(\Delta W^{(m)}u)_m\|_F^2 + \sum_{k \neq m} \|(\Delta W^{(m)}u)_k\|_F^2. \tag{24}$$

We control the first term and the second in the right hand of the equation respectively.

**Step 1:** For the first term:

$$(\Delta W^{(m)}u)_m = \sum_{k=1}^n W_{mk}u_k.$$

Express the $(i, j)$ element of $(\Delta W^{(m)}u)_m$ in the following form:

$$(\Delta W^{(m)}u)_m^{ij} = \sum_{k=1}^n (W_{mk}u_k)^{ij} = \sum_{k=1}^n \sum_{s=1}^{d_k} W_{mk}^{is}u_k^{sj}.$$

For a fixed pair of $(i, j)$, $W_{mk}^{is}$ on the right hand side of the equality are a row of $W$. Therefore, these $W_{mk}^{is}$ are mean zero , independent, standard sub-Gaussian random variables. By Hoeffding inequality, we have

$$P(|\sum_{k=1}^n \sum_{s=1}^{d_k} W_{mk}^{is}u_k^{sj}| \geq t) \leq 2exp(\frac{-ct^2}{\sum_{k=1}^n \sum_{s=1}^{d_k}(u_k^{sj})^2}) \leq 2exp(\frac{-ct^2}{\|u\|_F^2}) = 2exp(\frac{-ct^2}{nr}),$$

where $c$ is an absolute constant. Let $t = C_1\sqrt{n\log n}$ with $cC_1^2 \geq 5r$, we deduce that with probability at least $1 - 2n^{-5}$,

$$|\sum_{k=1}^n \sum_{s=1}^{d_k} W_{mk}^{is}u_k^{sj}| \leq C_1\sqrt{n\log n}.$$

By taking a union bound over $(i, j)$, we deduce that with probability at least $1 - 2d_m r n^{-5}$,

$$\forall (i, j) \in [d_m] \times [r], \quad |(\Delta W^{(m)} u)_m^{ij}| \leq C_1 \sqrt{n \log n}.$$

As a result, with probability at least $1 - 2d_m r n^{-5}$,

$$\|(\Delta W^{(m)} u)_m\|_F = \sqrt{\sum_{i=1}^{d_m} \sum_{j=1}^{r} ((\Delta W^{(m)} u)_m^{ij})^2} \leq \sqrt{d_m r} C_1 \sqrt{n \log n}. \tag{25}$$

**Step 2:** For the second term:

$$\begin{aligned}
\sum_{k \neq m} \|(\Delta W^{(m)} u)_k\|_F^2 &= \sum_{k \neq m} \|W_{km} u_m\|_F^2 \\
&\leq \|u_m\|_F^2 \sum_{k \neq m} \|W_{km}\|_F^2 \\
&\leq M_m^2 \sum_{k \neq m} \|W_{km}\|_F^2 \\
&= M_m^2 \sum_{k \neq m} \sum_{(i,j) \in [d_k] \times [d_m]} (W_{km}^{ij})^2.
\end{aligned} \tag{26}$$

$W_{km}^{ij}$ on the right hand of the last equality are $(nd - d_m)d_m$ mean zero, independent, standard sub-Gaussian random variables. Therefore, $((W_{km}^{ij})^2$ are $(nd - d_m)d_m$ independent sub-exponential random variables. Let $\mu = E[(W_{km}^{ij})^2]$, by Bernstein inequality:

$$P(|\sum_{k \neq m} \sum_{(i,j) \in [d_k] \times [d_m]} [(W_{km}^{ij})^2 - \mu]| \geq t) \leq 2 exp(-c \min(\frac{t^2}{(nd - d_m)d_m}, t)),$$

where $c$ is an absolute constant. Taking $t = cndd_m$, we derive that with probability at least $1 - 2exp(-cndd_m)$,

$$|\sum_{k \neq m} \sum_{(i,j) \in [d_k] \times [d_m]} [(W_{km}^{ij})^2 - \mu]| \leq ndd_m.$$

Notice that $\mu = \|W_{km}^{ij}\|_{L_2}^2 \leq (C' \|W_{km}^{ij}\|_{\psi_2} \sqrt{2})^2 = \mu'$, where $\mu'$ is an absolute constant. With probability at least $1 - 2exp(-cndd_m)$,

$$\sum_{k \neq m} \sum_{(i,j) \in [d_k] \times [d_m]} (W_{km}^{ij})^2 \leq |\sum_{k \neq m} \sum_{(i,j) \in [d_k] \times [d_m]} [(W_{km}^{ij})^2 - \mu]| + (nd - d_m)d_m \mu \leq (\mu' + 1)ndd_m. \tag{27}$$

Combining (24),(25),(26) and (27), we derive that with probability at least $1 - 2exp(-cndd_m) - 2d_m r n^{-5}$,

$$\begin{aligned}
\|\Delta W^{(m)} u\|_F &\leq \|(\Delta W^{(m)} u)_m\|_F + \sqrt{\sum_{k \neq m} \|(\Delta W^{(m)} u)_k\|_F^2} \\
&\leq \sqrt{d_m r} C_1 \sqrt{n \log n} + \sqrt{\mu' + 1} \sqrt{ndd_m} M_m
\end{aligned}$$

Finally, take a union bound over $u \in U^{(m)}$ and rewrite the constant as $C_1 = \max\{\sqrt{d_{max} r} C_1, \sqrt{(\mu' + 1)dd_{max}}\}$, where $d_{max} = \max_{i \in [n]} d_i$. With probability at least $1 - 6n^2 exp(-cndd_m) - 6d_m r n^{-3}$:

$$\|\Delta W^{(m)} u\|_F \leq C_1 \sqrt{n \log n} + C_1 \sqrt{n} M_m,$$

which completes the proof. $\square$

## B. Proof of Theorem 3.3

**Lemma B.1** (Davis-Kahn). *let $A, E \in \mathbb{R}^{n \times n}$ be symmetrical matrices, $\widetilde{A} = A + E$, $\delta = \lambda_d(A) - \lambda_{d+1}(A)$, where $1 \leq d \leq n - 1$. let $U, \widetilde{U}$ be the top $d$ leading eigenvectors of $A$ and $\widetilde{A}$ with $\|U\|_F = \|\widetilde{U}\|_F = \sqrt{nd}$, then the following holds:*

$$d_F(U, \widetilde{U}) \leq \frac{\sqrt{2}\|EU\|_F}{\delta - \|E\|_2}. \tag{28}$$

**Lemma B.2** (Weyl Inequality). *Let $A, E \in \mathbb{R}^{n \times n}$ be symmetrical matrices. $\lambda_1 \geq \lambda_2 \geq \cdots \geq \lambda_n$ are the eigenvalues of $A$ and $\widetilde{\lambda_1} \geq \widetilde{\lambda_2} \geq \cdots \widetilde{\lambda_n}$ are the eigenvalues of $\widetilde{A} = A + E$. Then, $|\widetilde{\lambda_i} - \lambda_i| \leq \|E\|_2, \quad \forall i \in [n]$*

*Proof of Theorem 3.3 .* We claim that the top $r$ eigenvalues of $G^\star G^{\star\top}$ are all $n$ and other eigenvalues of $G^\star G^{\star\top}$ are 0; moreover, columns of $G^\star$ are $r$ linearly independent eigenvectors of $G^\star G^{\star\top}$ corresponding to eigenvalue $n$.

Now, we prove the claim. Since $G^\star \in O_{d,r}^n$, we have

$$G^\star G^{\star\top} G^\star = G^\star n I_r = n G^\star.$$

This implies that columns of $G^\star$ are eigenvectors of $G^\star G^{\star\top}$ corresponding to the eigenvalue $n$. By the rank inequality $\text{rank}(AB) \leq \min\{\text{rank}(A), \text{rank}(B)\}$, we have

$$r = \text{rank}(G^{\star\top} G^\star) \leq \text{rank}(G^\star) \leq r.$$

This implies that the columns of $G^\star$ are linearly independent and the algebraic multiplicity of $n$ is at least $r$. By the rank inequality $\text{rank}(AB) \geq \text{rank}(A) + \text{rank}(B) - r$ we derive,

$$r = \text{rank}(G^\star) + \text{rank}(G^{\star\top}) - r \leq \text{rank}(G^\star G^{\star\top}) \leq \text{rank}(G^\star) = r$$

This implies $\text{rank}(G^\star G^{\star\top}) = r$, so $G^\star G^{\star\top}$ has at most $r$ non-zero eigenvalues. Combining the above discussion, the $r$ non-zero eigenvalues must be $n$. This finishes the proof of the claim. Next, we prove the theorem in two steps.

**Step 1:** Prove $d_F(\widetilde{G}, G^\star) \leq C_0 \sigma$, using standard matrix permutation bounds.

With probability at least $1 - O(n^{-2})$, the bounds in lemma A.1 hold. We choose $c_0$ such that $c_0 C_2 < \frac{1}{8}$. Using Davis-Kahn inequality we get

$$\begin{aligned}
d_F(\sqrt{d}G^\star, \sqrt{d}\widetilde{G}) &\leq \frac{\sqrt{2}\|\sigma W \sqrt{d} G^\star\|_F}{n - \|\sigma W\|_2} \\
&\leq \frac{\sqrt{2d}\sigma \|W\|_2 \|G^\star\|_F}{n - \sigma\|W\|_2} \\
&\leq \frac{\sqrt{2d}\sigma C_2 \sqrt{n}\sqrt{nr}}{n - c_0\sqrt{n}C_2\sqrt{n}} \\
&= \frac{8\sqrt{2dr}C_2\sigma}{7}.
\end{aligned}$$

Dividing $\sqrt{d}$ on both sides, we get

$$d_F(G^\star, \widetilde{G}) \leq \frac{8\sqrt{2r}C_2\sigma}{7}$$

**Step 2:** Prove $d_\infty(\widetilde{G}, G^\star) \leq C_0 \sigma \sqrt{\log n/n}$, using leave-one-out technique.

Let $\lambda_1 \geq \lambda_2 \geq \cdots \geq \lambda_r$ be the largest $r$ eigenvalues of $C$ and let $\Lambda = diag(\lambda_1, \cdots, \lambda_r)$. Using Weyl inequality, we get

$$|\lambda_i - n| \leq \sigma\|W\|_2 \leq c_0\sqrt{n}C_2\sqrt{n} \leq \frac{1}{8}n$$

$$\frac{7}{8}n \leq \lambda_i \leq \frac{9}{8}n, \quad \forall i \in [r].$$

Therefore $\Lambda$ is invertible. Since columns of $\widetilde{G}$ are eigenvectors of $C$, we have

$$\widetilde{G}\Lambda = C\widetilde{G} = G^\star G^{\star\top}\widetilde{G} + \sigma W\widetilde{G},$$
$$\widetilde{G} = G^\star G^{\star\top}\widetilde{G}\Lambda^{-1} + \sigma W\widetilde{G}\Lambda^{-1}.$$

Assume $d_F(G^\star, \widetilde{G}) = \|\widetilde{G} - G^\star Q\|_F, \quad Q \in O(r)$, Then, $\forall m \in [n]$,

$$
\begin{aligned}
\|(\widetilde{G} - G^\star Q)_m\|_F &= \|G_m(G^{\star\top}\widetilde{G}\Lambda^{-1} - Q) + \sigma(W\widetilde{G})_m\Lambda^{-1}\|_F \\
&\leq \|G_m^\star\|_F\|G^{\star\top}\widetilde{G} - Q\Lambda\|_2\|\Lambda^{-1}\|_2 + \sigma\|(W\widetilde{G})_m\|_F\|\Lambda^{-1}\|_2 \\
&\leq \frac{\sqrt{r}\|G^{\star\top}\widetilde{G} - Q\Lambda\|_2}{\frac{7}{8}n} + \frac{\sigma\|W\widetilde{G}\|_\infty}{\frac{7}{8}n}.
\end{aligned}
\tag{29}
$$

We bound the two terms in the last inequality respectively.

We first bound $\|G^{\star\top}\widetilde{G} - Q\Lambda\|_2$.

$$
\begin{aligned}
\|G^{\star\top}\widetilde{G} - Q\Lambda\|_2 &\leq \|G^{\star\top}\widetilde{G} - nQ\|_F + \|nQ - Q\Lambda\|_2 \\
&= \|G^{\star\top}(\widetilde{G} - G^\star Q)\|_F + \|nI_r - \Lambda\|_2 \\
&\leq \sqrt{nr}d_F(\widetilde{G}, G^\star) + C_2\sigma\sqrt{n} \\
&\leq \frac{8\sqrt{2}rC_2\sigma\sqrt{n}}{7} + C_2\sigma\sqrt{n} \\
&= (\frac{8\sqrt{2}rC_2}{7} + C_2)\sigma\sqrt{n}.
\end{aligned}
\tag{30}
$$

Next, We bound $\|W\widetilde{G}\|_\infty$.

$$\|W\widetilde{G}\|_\infty = \max_{m \in [n]} \|w_m^\top\widetilde{G}\|_F, \tag{31}$$

where $w_m \in \mathbb{R}^{nd \times d_i}$ is the $m$-th block column of $W$. Take an arbitrary $m \in [n]$. Let $\widetilde{G}^{(m)}$ be a $nd \times r$ matrix, whose columns are the top $r$ leading eigenvectors of $C^{(m)}$, with $\|\widetilde{G}^{(m)}\|_F = \sqrt{nr}$. Assuming that $d_F(\widetilde{G}, \widetilde{G}^{(m)}) = \|\widetilde{G} - \widetilde{G}^{(m)}Q\|_F, \ Q \in O(r)$, we have

$$
\begin{aligned}
\|w_m^\top\widetilde{G}\|_F &\leq \|w_m^\top\widetilde{G}^{(m)}Q\|_F + \|w_m^\top(\widetilde{G} - \widetilde{G}^{(m)}Q)\|_F \\
&\leq \|w_m^\top\widetilde{G}^{(m)}\|_F + \|w_m\|_2 d_F(\widetilde{G}^{(m)}, \widetilde{G}).
\end{aligned}
\tag{32}
$$

Using wely inequality, we have

$$|\lambda_r(C^{(m)}) - n| \leq \sigma\|W^{(m)}\|_2 \leq c_0 C_2 n \leq \frac{1}{8}n,$$
$$|\lambda_{r+1}(C^{(m)}) - 0| \leq \sigma\|W^{(m)}\|_2 \leq c_0 C_2 n \leq \frac{1}{8}n.$$

Therefore $\delta(C^{(m)}) = \lambda_r(C^{(m)}) - \lambda_{r+1}(C^{(m)}) \geq \frac{7}{8}n - \frac{1}{8}n = \frac{3}{4}n$. Using Davis-Kahn inequality, we deduce

$$
\begin{aligned}
d_F(\sqrt{d}\widetilde{G}^{(m)}, \sqrt{d}\widetilde{G}) &\leq \frac{\sqrt{2}\|\sigma\Delta W^{(m)}\sqrt{d}\widetilde{G}^{(m)}\|_F}{\delta(C^{(m)}) - \|\sigma\Delta W^{(m)}\|_2} \\
&\leq \frac{\sqrt{2d}\sigma\|\Delta W^{(m)}\widetilde{G}^{(m)}\|_F}{\frac{3}{4}n - \frac{1}{8}n} \\
&= \frac{8\sqrt{2d}\sigma\|\Delta W^{(m)}\widetilde{G}^{(m)}\|_F}{5n}.
\end{aligned}
\tag{33}
$$

Combining (33) and (32) we have

$$\|w_m^\top \widetilde{G}\|_F \leq \|w_m^\top \widetilde{G}^{(m)}\|_F + C_2\sqrt{n}\frac{8\sqrt{2}\sigma\|\Delta W^{(m)}\widetilde{G}^{(m)}\|_F}{5n}$$

$$\leq \|\Delta W^{(m)}\widetilde{G}^{(m)}\|_F + \frac{8\sqrt{2}C_2 c_0\|\Delta W^{(m)}\widetilde{G}^{(m)}\|_F}{5} \tag{34}$$

$$\leq (\frac{\sqrt{2}}{5} + 1)\|\Delta W^{(m)}\widetilde{G}^{(m)}\|_F,$$

where the second inequality is because $w_m^\top\widetilde{G}$ is a block in $\Delta W^{(m)}\widetilde{G}^{(m)}$. Noticing that $\widetilde{G}^{(m)}$ is independent from $\Delta W^{(m)}$, we choose $U^{(m)} = \{\widetilde{G}^{(m)}\}$ in lemma 3.4. With probability at least $1 - O(n^{-2})$,

$$\forall 1 \leq m \leq n, \quad \|\Delta W^{(m)}\widetilde{G}^{(m)}\|_F \leq C_1\sqrt{n\log n} + C_1\sqrt{n}\|\widetilde{G}_m^{(m)}\|_F, \tag{35}$$

where $\widetilde{G}_m^{(m)}$ is the $m$-th $d \times r$ block of $\widetilde{G}^{(m)}$. Combining (31),(34) and (35) we derive

$$\|W\widetilde{G}\|_\infty \leq (\frac{\sqrt{2}}{5} + 1)(C_1\sqrt{n\log n} + C_1\sqrt{n}\max_{m\in[n]}\|\widetilde{G}_m^{(m)}\|_F). \tag{36}$$

Next, we will prove that $\|\widetilde{G}_m^{(m)}\|_F \leq \frac{8}{7}r$. Since the columns of $\widetilde{G}^{(m)}$ are leading eigenvectors of $C^{(m)}$, we have

$$\widetilde{G}^{(m)}\widetilde{\Lambda} = C^{(m)}\widetilde{G}^{(m)} = G^\star G^{\star\top}\widetilde{G}^{(m)} + \sigma W^{(m)}\widetilde{G}^{(m)},$$

where $\widetilde{\Lambda}$ is a $r \times r$ diagonal matrix whose diagonal entries are the top $r$ largest eigenvalues of $C^{(m)}$. Using wely inequality, we deduce that all elements of $\widetilde{\Lambda}$ is larger than $\frac{7}{8}n$. Therefore,$\forall m \in [n]$,

$$\|\widetilde{G}_m^{(m)}\|_F = \| \left( G_m^\star G^{\star\top}\widetilde{G}^{(m)} + \sigma(W^{(m)}\widetilde{G}^{(m)})_m \right) \widetilde{\Lambda}^{-1}\|_F$$

$$= \|G_m^\star G^{\star\top}\widetilde{G}^{(m)}\widetilde{\Lambda}^{-1}\|_F$$

$$\leq \|G_m^\star\|_2\|G^{\star\top}\|_F\|\widetilde{G}^{(m)}\|_F\|\widetilde{\Lambda}^{-1}\|_2 \tag{37}$$

$$\leq \frac{8}{7}r,$$

where the second equality is because that the $m$-th block row of $W^{(m)}$ is 0. Combining (36) and (37), we get

$$\|W\widetilde{G}\|_\infty \leq (\frac{\sqrt{2}}{5} + 1)(C_1\sqrt{n\log n} + \frac{8}{7}r^{\frac{3}{2}}C_1\sqrt{n}). \tag{38}$$

Finally, combining (29), (30) and (38) we get

$$\|(\widetilde{G} - G^\star Q)_m\|_F \leq \frac{\sqrt{r}(\frac{8\sqrt{2}rC_2}{7} + C_2)\sigma\sqrt{n}}{\frac{7}{8}n} + \frac{\sigma(\frac{\sqrt{2}}{5} + 1)(C_1\sqrt{n\log n} + \frac{8}{7}r^{\frac{3}{2}}C_1\sqrt{n})}{\frac{7}{8}n}, \tag{39}$$

which completes the proof.

$\square$

## C. Proof of Lemma 3.5

**Lemma C.1.** *Let $A \in \mathbb{R}^{m\times m}$ with SVD decomposition $A = UDV^\top$ then,*

$$\|A\|_\star = \max_{Q\in O(m)} \langle Q, A\rangle = \langle UV^\top, A\rangle, \tag{40}$$

*Proof.* Let $Q \in O(m)$. We use $\sigma_i(A)$ to denote the $i$-th singular value of $A$ and $\sigma_i(Q)$ to denote the $i$-th singular value of $Q$. Using von Neumann's inequality, we have

$$\langle Q, A \rangle \leq \sum_{i=1}^{n} \sigma_i(Q)\sigma_i(A) \leq \sum_{i=1}^{n} \sigma_i(A) = \|A\|_\star, \tag{41}$$

where the second inequality is because $|\sigma_i(Q)| = 1$. On the other hand, let $Q = UV^\top$, we have

$$\langle Q, A \rangle = tr(A^\top Q) = tr(VDV^\top) = tr(D) = \|A\|_\star. \tag{42}$$

Therefore (40) holds. $\qquad\square$

**Proposition C.2.** *Let* $G_1, G_2 \in O_{d,r}^n$ *with* $d_F(G_1, G_2) \leq \epsilon\sqrt{nr}$, *then* $\forall i \in [n]$,

$$(1 - \frac{\epsilon^2 r}{2})n \leq \sigma_i(G_1^\top G_2) \leq n. \tag{43}$$

*Proof.* Since $d_F(G_1, G_2) \leq \epsilon\sqrt{nr}$, we have

$$\epsilon^2 nr \geq d_F^2(G_1, G_2) = \|G_1\|_F^2 + \|G_2\|_F^2 - 2\|G_1^\top G_2\|_\star$$
$$= 2\sum_{i=1}^{r}(n - \sigma_i(G_1^\top G_2)) \geq 2(n - \sigma_r(G_1^\top G_2)), \tag{44}$$

which implies

$$\sigma_r(G_1^\top G_2) \geq (1 - \frac{\epsilon^2 r}{2})n. \tag{45}$$

On the other hand, $\forall i \in [r]$,

$$\sigma_i(G_1^\top G_2) \leq \|G_1^\top G_2\|_2 \leq \|G_1\|_2 \|G_2\|_2 = n. \tag{46}$$

$\qquad\square$

**Lemma C.3.** *Let* $L = \frac{1}{n}C$, $x, y \in O_{d,r}^n$, $d_F(x, G^\star) \leq \epsilon\sqrt{n}$, $d_F(y, G^\star) \leq \epsilon\sqrt{n}$, *then*

$$d_F(Lx, Ly) \leq (2\epsilon + \sigma\frac{\|W\|_2}{n})d_F(x, y) \tag{47}$$

*Proof.* Let $d_F(x, y) = \|x - yQ\|_F$, we claim that

$$\langle y^\top x, Q \rangle = \|y^\top x\|_\star. \tag{48}$$

The reason is as follows: $Q = argmin_{P \in O(r)}\|x - yP\|_F^2 = argmax_{P \in O(r)}\langle x, yP \rangle = argmax_{P \in O(r)}\langle y^\top x, P \rangle$. By lemma C.1, $\langle y^\top x, Q \rangle = \max_{Q \in O(r)}\langle Q, y^\top x \rangle = \|y^\top x\|_\star$.

We now prove lemma C.3.

$$d_F(Lx, Ly) \leq \|Lx - LyQ\|_F$$
$$= \|\frac{1}{n}(G^\star G^{\star\top} + \sigma W)(x - yQ)\|_F$$
$$\leq \frac{1}{n}\|G^\star\|_2\|G^{\star\top}(x - yQ)\|_F + \frac{\sigma}{n}\|W(x - yQ)\|_F \tag{49}$$
$$\leq \frac{\|G^{\star\top}(x - yQ)\|_F}{\sqrt{n}} + \frac{\sigma\|W\|_2}{n}d_F(x, y).$$

The second term on the right of the last inequality is exactly what we wanted and we only needs to bound the first term $\|G^{\star\top}(x - yQ)\|_F$. Letting $d_F(y, G^\star) = \|y - G^\star Q_y\|_F$, we have

$$
\begin{aligned}
\|G^{\star\top}(x - yQ)\|_F &\leq \|(y - G^\star Q_y)^\top(x - yQ)\|_F + \|y^\top(x - yQ)\|_F \\
&\leq d_F(y, G^\star)d_F(x, y) + \|y^\top(x - yQ)\|_F \\
&\leq \epsilon\sqrt{n}d_F(x, y) + \|y^\top x - nQ)\|_F.
\end{aligned}
\tag{50}
$$

Using $\sigma_i(y^\top x)$ to denote the $i$-th singular value of $y^\top x$, we have

$$
\begin{aligned}
\|y^\top x - nQ)\|_F^2 &= \|y^\top x\|_F^2 + n^2 r - 2n < Q, y^\top x > \\
&= \sum_{i=1}^r (\sigma_i(y^\top x))^2 + n^2 r - 2n \sum_{i=1}^r \sigma_i(y^\top x) \\
&= \sum_{i=1}^r (n - \sigma_i(y^\top x))^2,
\end{aligned}
\tag{51}
$$

where the second equality comes from (48). Since $d_F(x, y) \leq d_F(x, G^\star) + d_F(y, G^\star) \leq 2\epsilon\sqrt{n}$, using proposition (C.2), we have

$$
(1 - 2\epsilon^2)n \leq \sigma_i(y^\top x) \leq n.
\tag{52}
$$

Combining (51),(52), we have

$$
\begin{aligned}
\|y^\top x - nQ)\|_F^2 &= \sum_{i=1}^r (n - \sigma_i(y^\top x))^2 \\
&\leq (n - \sigma_r(y^\top x)) \sum_{i=1}^r (n - \sigma_i(y^\top x)) \\
&\leq 2\epsilon^2 n(nr - \|y^\top x\|_\star) \\
&= \epsilon^2 n d_F^2(x, y).
\end{aligned}
\tag{53}
$$

Combining (49),(50), (53), we have

$$
\begin{aligned}
d_F(Lx, Ly) &\leq \frac{\|G^{\star\top}(x - yQ)\|_F}{\sqrt{n}} + \frac{\sigma\|W\|_2}{n}d_F(x, y) \\
&\leq 2\epsilon d_F(x, y) + \frac{\sigma\|W\|_2}{n}d_F(x, y) \\
&= (2\epsilon + \frac{\sigma\|W\|_2}{n})d_F(x, y).
\end{aligned}
\tag{54}
$$

$\square$

**Lemma C.4** (Davis Kahn)**.** *Let* $X, X_E$ *be Hermitian matrices with* $X_E = X + E$. $\psi$ *and* $\psi_E$ *are the top* $d$ *leading eigenvectors of* $X$ *and* $X_E$ *with* $\psi^\top\psi = I_d = \psi_E^\top\psi_E$, *then*

$$
d_F(\psi, \psi_E) \leq \frac{\sqrt{2}\|E\psi_E\|_F}{\lambda_d(X) - \lambda_{d+1}(X_E)}.
\tag{55}
$$

*Proof.* This is theorem 4.9 in (Ling, 2022b) $\square$

**Lemma C.5.** *Let* $X, Y \in \mathbb{R}^{d \times r}$, $\sigma_r(X) > 0$, *then*

$$
\|\pi(X) - \pi(Y)\|_F \leq \frac{4\|X - Y\|_F}{\sigma_r(X)},
\tag{56}
$$

*where* $\pi$ *is the projection operator on stiefel manifold.*

*Proof.* Using SVD decomposition, $X = U_X D_X V_X^\top$, where $U_X \in \mathbb{R}^{d \times r}$ satisfies $U_X^\top U_X = I_r$, $D_X = diag(\sigma_1(X), \cdots \sigma_r(X))$, $V_X \in O(r)$. Similarly we have the SVD decomposition $Y = U_Y D_Y V_Y^\top$. We construct two auxiliary matrices as follows:

$$\widetilde{X} = \begin{pmatrix} O_{d \times d} & X \\ X^\top & O_{r \times r} \end{pmatrix}, \quad \widetilde{Y} = \begin{pmatrix} O_{d \times d} & Y \\ Y^\top & O_{r \times r} \end{pmatrix}.$$

Notice:

$$|\lambda I - \widetilde{X}| = \begin{vmatrix} \lambda I_d & -X \\ -X^\top & \lambda I_r \end{vmatrix} = \begin{vmatrix} \lambda I_d & -X \\ O & \lambda I_r - \frac{1}{\lambda} X^\top X \end{vmatrix}$$

$$= \lambda^{d-r} |\lambda^2 I_r - X^\top X|$$

$$= \lambda^{d-r} (\lambda^2 - \sigma_1(X)^2)(\lambda^2 - \sigma_2(X)^2) \cdots (\lambda^2 - \sigma_r(X)^2).$$

We deduce that the eigenvalues of $\widetilde{X}$ are $\sigma_1(X), \cdots, \sigma_r(X), 0$ with multiplicity $d - r$, and $-\sigma_1(X), \cdots -\sigma_r(X)$. Similarly, the eigenvalues of $\widetilde{Y}$ are $\sigma_1(Y), \cdots, \sigma_r(Y), 0$ with multiplicity $d - r$, and $-\sigma_1(Y), \cdots -\sigma_r(Y)$.

Define

$$M_X = \frac{1}{\sqrt{2}} \begin{pmatrix} U_X \\ V_X \end{pmatrix}, \quad M_Y = \frac{1}{\sqrt{2}} \begin{pmatrix} U_Y \\ V_Y \end{pmatrix}.$$

. It is obvious that $M_X^\top M_X = M_Y^\top M_Y = I_r$. Besides, we have $\widetilde{X} M_X = M_X D_X$ and $\widetilde{Y} M_Y = M_Y D_Y$. Thus, $M_X$ and $M_Y$ are the top $r$ leading eigenvectors of $\widetilde{X}$ and $\widetilde{Y}$ respectively. Using Davis Kahn inequality we get

$$d_F(M_X, M_Y) \leq \frac{\sqrt{2} \|(\widetilde{Y} - \widetilde{X}) M_Y\|_F}{\sigma_r(X)} \leq \frac{\sqrt{2} \|(\widetilde{Y} - \widetilde{X})\|_F \|M_Y\|_2}{\sigma_r(X)} = \frac{2 \|(Y - X)\|_F}{\sigma_r(X)}.$$

From above we deduce that there exists $Q \in O(r)$ such that

$$\left\| \begin{pmatrix} U_X - U_Y Q \\ V_X - V_Y Q \end{pmatrix} \right\|_F = \sqrt{\|U_X - U_Y Q\|_F^2 + \|V_X - V_Y Q\|_F^2} \leq \frac{2\sqrt{2} \|(Y - X)\|_F}{\sigma_r(X)}.$$

On the other hand,

$$\|\pi(X) - \pi(Y)\|_F = \|U_X V_X^\top - U_Y V_Y^\top\|_F$$

$$= \|U_X V_X^\top - U_Y Q V_X^\top\|_F + \|U_Y Q V_X^\top - U_Y V_Y^\top\|_F$$

$$\leq \|U_X - U_Y Q\|_F + \|V_X - V_Y Q\|_F$$

$$\leq \sqrt{2} \sqrt{\|U_X - U_Y Q\|_F^2 + \|V_X - V_Y Q\|_F^2}$$

$$= \frac{4 \|(Y - X)\|_F}{\sigma_r(X)}.$$

$\square$

**Corollary C.6.** *Let* $X, Y \in \mathbb{R}^{nd \times r}$ *with* $\min_{i \in [n]} \sigma_r(X_i) \geq \epsilon > 0$, *then*

$$d_F(\pi^n(X), \pi^n(Y)) \leq \frac{4 d_F(X, Y)}{\epsilon} \tag{57}$$

*Proof.* Assume that $d_F(X, Y) = \|X - YQ\|_F$.

$$d_F(\pi^n(X), \pi^n(Y)) \leq \|\pi^n(X) - \pi^n(Y)Q\|_F = \sqrt{\sum_{i=1}^n \|\pi(X_i) - \pi(Y_i Q)\|_F^2}$$

$$\leq \frac{4}{\epsilon} \sqrt{\sum_{i=1}^n \|X_i - Y_i Q\|_F^2} = \frac{4}{\epsilon} d_F(X, Y).$$

$\square$

**Lemma C.7** (Weyl)**.** *Let* $A, B$ *be* $m \times n$ *matrices, then for any* $i, j$ *with* $1 \le i, j, i + j - 1 \le \min(m, n)$, *the following inequality holds:*

$$\sigma_{i+j-1}(A + B) \le \sigma_i(A) + \sigma_j(B). \tag{58}$$

*Proof of Lemma 3.5.* Let $L = \frac{1}{n}C, L' = \frac{1}{n}C'$. In order to use corollary C.6 , we first provide a lower bound for $\sigma_r((Lx)_i)$ and $\sigma_r((L'y)_i)$.

Taking $i = r, j = 1$ in lemma C.7 we have,

$$\sigma_r(A) \ge \sigma_r(A + B) - \sigma_1(B). \tag{59}$$

Letting $A = (Lx)_i$, $B = -\frac{\sigma}{n}(Wx)_i$ and $A + B = (Lx)_i - \frac{\sigma}{n}(Wx)_i = \frac{G_i^\star G^{\star\top} x}{n}$, we derive

$$\sigma_r((Lx)_i) \ge \frac{1}{n}\sigma_r(G_i^\star G^{\star\top} x) - \frac{\sigma}{n}\sigma_1((Wx)_i) = \frac{1}{n}\sigma_r(G_i^\star G^{\star\top} x) - \frac{\sigma}{n}\|Wx\|_\infty. \tag{60}$$

Since $d_F(x, G^\star) \le \epsilon_1 \sqrt{nr}$, by proposition C.2, we have $\sigma_r(G^{\star\top} x) \ge (1 - \frac{\epsilon_1^2 r}{2})n$. Therefore, $\sigma_r((Lx)_i) \ge 1 - \frac{\epsilon_1^2 r}{2} - \sigma k_1 \sqrt{\log n / n}$. Similarly, we have $\sigma_r((L'y)_i) \ge 1 - \frac{\epsilon_2^2 r}{2} - \sigma k_2 \sqrt{\log n / n}$. Using corollary C.6 we derive

$$d_F(\pi^n(Lx), \pi^n(Ly)) \le \frac{4}{1 - \epsilon}d_F(Lx, Ly). \tag{61}$$

If $L = L'$, using lemma C.3 we have:

$$d_F(Lx, Ly) \le (2 \max\{\epsilon_1, \epsilon_2\}\sqrt{r} + \frac{\sigma\|W\|_2}{n})d_F(x, y). \tag{62}$$

Combining (61), (62), we complete the proof. □

# D. Proof of Proposition 3.6

With probability at least $1 - O(n^{-2})$, the results in Theorem 3.3 holds.

**Step 1:** $d_F(G^0, G^\star) \lesssim \sigma$.

$$d_F(G^0, G^\star) = d_F(\pi^n(\widetilde{G}), \pi^n(G^\star)) \le 4d_F(\widetilde{G}, G^\star) \le 4C_0\sigma, \tag{63}$$

where the first inequality is because of corollary C.6 and $\sigma_r(G_i^\star) = 1$.

**Step 2:** $d_\infty(G^0, G^\star) \lesssim \sigma\sqrt{n/\log n}$.

Assume that $d_\infty(\widetilde{G}, G^\star) = \|\widetilde{G} - G^\star Q\|_\infty$, then $\forall i$

$$\|G_i^0 - G_i^\star Q\|_F = \|\pi(\widetilde{G}_i) - \pi(G_i^\star Q)\|_F \le 4\|\widetilde{G}_i - G_i^\star Q\|_F \le 4d_\infty(\widetilde{G}, G^\star). \tag{64}$$

Therefore, $d_\infty(G^0, G^\star) \le \|G^0 - G^\star Q\|_\infty \le 4d_\infty(G^\star, \widetilde{G}) \lesssim \sigma\sqrt{\log n / n}$.

# E. Proof of Theorem 3.7

We split the proof into two parts: Theorem E.1 handles the initial point $G^0, G^{0,m}$ and Theorem E.2 handles the rest of points by induction.

**Theorem E.1.** *Let* $n \ge 2$, $\sigma \le \min\{\frac{\sqrt{n/r}}{12000\sqrt{2}C_2}, \frac{1}{24000k_2}\sqrt{\frac{n/r}{\log n}}, C_3\sqrt{n/\log n}\}$, *then with probability at least* $1 - O(n^{-2})$, *we have*

$$d_F(G^0, G^\star) \le k_3\sqrt{n}, \tag{65}$$

$$d_F(G^{0,m}, G^0) \le k_1, \tag{66}$$

$$\|WG^0\|_\infty \le k_2\sqrt{n\log n}, \tag{67}$$

*where* $C_2, C_1$ *are constants in lemma A.1 and lemma 3.4 respectively.* $c_0$ *and* $C_0$ *are constants in theorem 3.3.* $C_3 \le c_0$ *and* $C_3C_0 \le \frac{1}{2}$. $k_1 = k_3 = \frac{1}{600}, k_2 = 4C_1\sqrt{r} + 2C_2k_1$.

*Proof.* **Step 1:** $d_F(G^0, G^\star) \leq k_3 \sqrt{n}$.

Notice that $\min_{i \in [n]} \sigma_r(G_i^\star) = 1$. Using corollary C.6 we have:

$$d_F(G^0, G^\star) = d_F(\pi^n \widetilde{G}, \pi^n G^\star) \leq 4 d_F(\widetilde{G}, G^\star). \tag{68}$$

Using Davis Kahn inequality, we derive

$$d_F(\widetilde{G}, G^\star) \leq \frac{\sqrt{2}\sigma \|W\|_2 \|G^\star\|_F}{n - \sigma \|W\|_2} \leq \frac{\sqrt{2}\sigma C_2 \sqrt{n}\sqrt{nr}}{n - \sigma C_2 \sqrt{n}} \leq \frac{1}{6000}\sqrt{n}. \tag{69}$$

Therefore, $d_F(G^0, G^\star) \leq \frac{4}{6000}\sqrt{n} \leq k_3 \sqrt{n}$.

**Step 2:** $d_F(G^{0,m}, G^0) \leq k_1$.

Since $\sigma \leq C_3 \sqrt{n/\log n}$, using theorem 3.3 we get

$$d_\infty(\widetilde{G}, G^\star) = \|\widetilde{G} - G^\star Q\|_\infty \leq C_0 C_3 \leq \frac{1}{2}. \tag{70}$$

Using lemma C.7 we have

$$\begin{aligned}
\sigma_r(\widetilde{G}_i) &\geq \sigma_r(G_i^\star) - \sigma_1(G_i^\star Q - \widetilde{G}_i) \\
&= 1 - \|\widetilde{G}_i - G_i^\star Q\|_2 \\
&\geq 1 - d_\infty(\widetilde{G}, G^\star) \geq \frac{1}{2}.
\end{aligned} \tag{71}$$

Therefore, $\min_{i \in [n]} \sigma_r(\widetilde{G}_i) \geq \frac{1}{2}$. Using corollary C.6, we derive

$$d_F(G^0, G^{0,m}) = d_F(\pi^n \widetilde{G}, \pi^n \widetilde{G}^{(m)}) \leq 8 d_F(\widetilde{G}, \widetilde{G}^{(m)}) \tag{72}$$

Using weyl inequality(lemma B.2):

$$\begin{aligned}
|\lambda_r(C^{(m)}) - n| &\leq \sigma \|W^{(m)}\|_2 \leq \sigma C_2 \sqrt{n}, \\
|\lambda_{r+1}(C^{(m)}) - 0| &\leq \sigma \|W^{(m)}\|_2 \leq \sigma C_2 \sqrt{n}.
\end{aligned} \tag{73}$$

Using Davis Kahn,

$$d_F(\widetilde{G}, \widetilde{G}^{(m)}) \leq \frac{\sqrt{2}\sigma \|\Delta W^{(m)} \widetilde{G}^{(m)}\|_F}{n - 3\sigma C_2 \sqrt{n}}. \tag{74}$$

Since $\widetilde{G}^{(m)}$ and $\Delta W^{(m)}$ are independent, we choose $U^{(m)} = \{\widetilde{G}^{(m)}\}$ in lemma 3.4, then

$$\forall m \in [n], \|\Delta W^{(m)} \widetilde{G}^{(m)}\|_F \leq C_1 \sqrt{n \log n} + C_1 \sqrt{n} \|\widetilde{G}_m^{(m)}\|_F \tag{75}$$

Notice that

$$C^{(m)} \widetilde{G}^{(m)} = G^\star G^{\star\top} \widetilde{G}^{(m)} + \sigma W^{(m)} \widetilde{G}^{(m)} = \widetilde{G}^{(m)} \Lambda, \tag{76}$$

where $\Lambda = diag(\lambda_1(C^m), \cdots, \lambda_r(C^m))$. $\forall m \in [n]$,

$$\|\widetilde{G}_m^{(m)}\| = \|G_m^\star G^{\star\top} \widetilde{G}^{(m)} \Lambda^{-1}\|_F \leq \|G_m^\star\|_2 \|G^{\star\top}\|_2 \|\widetilde{G}^{(m)}\|_F \|\Lambda^{-1}\|_2 \leq \sqrt{n}\sqrt{nr} \frac{1}{\lambda_r(C^{(m)})} \leq 2\sqrt{r}. \tag{77}$$

Combining (74),(75) and (77), we have

$$d_F(\widetilde{G}, \widetilde{G}^{(m)}) \leq \frac{\sqrt{2}\sigma \cdot 3\sqrt{r} C_1 \sqrt{n \log n}}{n - 3 \cdot \frac{n}{12000\sqrt{2}}} \leq \frac{\sqrt{2}}{12000}. \tag{78}$$

Substituting (78) into (72), we derive

$$d_F(G^0, G^{0,m}) \leq 8 \times \frac{\sqrt{2}}{12000} < k_1. \tag{79}$$

**Step 3:** $\|WG^0\|_\infty \le k_2\sqrt{n\log n}$.

Notice that $G^{0,m}$ is independent from $\Delta W^{(m)}$, we choose $U^{(m)} = \{G^{0,m}\}$ in lemma 3.4, then

$$\forall m \in [n], \|\Delta W^{(m)}G^{0,m}\|_F \le C_1\sqrt{n\log n} + C_1\sqrt{nr} \le 2C_1\sqrt{r}\sqrt{n\log n}. \tag{80}$$

$\forall m \in [n]$, let $d_F(G^0, G^{0,m}) = \|G^0 - G^{0,m}Q_m\|_F$, then

$$\begin{aligned}
\|W_m^\top G^0\|_F &= \|W_m^\top(G^0 - G^{0,m}Q_m) + W_m^\top G^{0,m}Q_m\|_F \\
&\le C_2\sqrt{n}d_F(G^0, G^{0,m}) + \|W_m^\top G^{0,m}\|_F \\
&\le C_2 k_1\sqrt{n} + \|\Delta W^{(m)}G^{0,m}\|_F \\
&\le C_2 k_1\sqrt{n} + 2C_1\sqrt{r}\sqrt{n\log n} \\
&\le k_2\sqrt{n\log n}.
\end{aligned} \tag{81}$$

Therefore, $\|WG^0\|_\infty = \max_{1\le m\le n}\|W_m^\top G^0\|_F \le k_2\sqrt{n\log n}$. $\qquad\square$

**Theorem E.2.** *Under the condition of theorem E.1, with probability at least $1 - O(n^{-2})$, $\forall t \in \{0, 1, \cdots 3n^2 - 1\}$, the following inequalities hold:*

$$d_F(G^t, G^\star) \le k_3\sqrt{n}, \tag{82}$$

$$d_F(G^{t,m}, G^t) \le k_1 \quad \forall m \in [n], \tag{83}$$

$$\|WG^t\|_\infty \le k_2\sqrt{n\log n}. \tag{84}$$

*Proof.* Let $T = 3n^2 - 1$. Choosing $U^{(m)} = \{G^{0,m}, G^{1,m}, \cdots, G^{T,m}\}$, we derive that with probability at least $1 - O(n^{-2})$,

$$\forall m \in [n], \max_{0\le t\le T}\|\Delta W^{(m)}G^{t,m}\|_F \le C_1\sqrt{n\log n} + C_1\sqrt{r}\sqrt{n} \le 2C_1\sqrt{r}\sqrt{n\log n}. \tag{85}$$

By theorem E.1, the initial points satisfies the properties in theorem E.2. We then prove the theorem by induction. Assuming that the three inequalities hold for some $t \in \{0, 1, \cdots, T - 1\}$, we prove the inequalities for $t + 1$.

**Step 1:** $d_F(G^{t+1}, G^\star) \le k_3\sqrt{n}$.

Let $L = \frac{1}{n}C$. Since $\min_{i\in[n]} \sigma_r(G_i^\star) = 1$, using corollary C.6 we get

$$d_F(G^{t+1}, G^\star) = d_F(\pi^n LG^t, \pi^n G^\star) \le 4d_F(LG^t, G^\star) \le 4\sqrt{n}d_\infty(LG^t, G^\star). \tag{86}$$

Then, we bound $d_\infty(LG^t, G^\star)$.

$$\begin{aligned}
d_\infty(LG^t, G^\star) &= \min_{Q\in O(r)} \|LG^t - G^\star Q\|_\infty \\
&= \min_{Q\in O(r)} \|\frac{1}{n}G^\star(G^{\star\top}G^t - nQ) + \frac{1}{n}\sigma WG^t\|_\infty \\
&\le \frac{1}{n}\min_{Q\in O(r)} \|G^\star(G^{\star\top}G^t - nQ)\|_\infty + \frac{1}{n}\sigma\|WG^t\|_\infty \\
&= \frac{1}{n}\min_{Q\in O(r)} \|G^{\star\top}G^t - nQ\|_F + \frac{1}{n}\sigma\|WG^t\|_\infty.
\end{aligned} \tag{87}$$

Since $\|WG^t\|_\infty$ is bounded by the induction assumption, we only needs to bound $\min_{Q\in O(r)}\|G^{\star\top}G^t - nQ\|_F$.

$$
\begin{aligned}
\min_{Q\in O(r)}\|G^{\star\top}G^t - nQ\|_F^2 &= \|G^{\star\top}G^t\|_F^2 + n^2 r - 2n\max_{Q\in O(r)}\langle Q, G^{\star\top}G^t\rangle\\
&= \|G^{\star\top}G^t\|_F^2 + n^2 r - 2n\|G^{\star\top}G^t\|_\star\\
&= \sum_{i=1}^r (n - \sigma_i(G^{\star\top}G^t))^2\\
&\leq (\sum_{i=1}^r n - \sigma_i(G^{\star\top}G^t))^2\\
&= (nr - \|G^{\star\top}G^t\|_\star)^2,
\end{aligned}
\tag{88}
$$

where the last inequality holds because $\sigma_i(G^{\star\top}G^t) \leq \|G^{\star\top}G^t\|_2 \leq n$. Notice that $d_F^2(G^\star, G^t) = \|G^\star\|_F^2 + \|G^t\|_F^2 - 2\|G^{\star\top}G^t\|_\star = 2nr - 2\|G^{\star\top}G^t\|_\star$. Combing the above expression with (87),(88) we derive that,

$$
\begin{aligned}
d_\infty(LG^t, G^\star) &\leq \frac{1}{n}(nr - \|G^{\star\top}G^t\|_\star) + \frac{1}{n}\sigma\|WG^t\|_\infty\\
&\leq \frac{1}{2n}d_F^2(G^\star, G^t) + \frac{1}{n}\sigma\|WG^t\|_\infty\\
&\leq \frac{k_3^2}{2} + \frac{\sigma k_2\sqrt{n\log n}}{n}\\
&\leq \frac{k_3^2}{2} + \frac{1}{24000}.
\end{aligned}
\tag{89}
$$

Substituting (89) into (86), we have

$$
d_F(G^{t+1}, G^\star) \leq (2k_3^2 + \frac{1}{6000})\sqrt{n} \leq k_3\sqrt{n}.
\tag{90}
$$

**Step 2:** $d_F(G^{t+1,m}, G^{t+1}) \leq k_1 \quad \forall m \in [n]$.

Let $L^{(m)} = \frac{1}{n}(G^\star G^{\star\top} + \sigma W^{(m)})$.

$$
d_F(G^{t+1,m}, G^{t+1}) = d_F(\pi^n L^{(m)}G^{t,m}, \pi^n LG^t) \leq d_F(\pi^n L^{(m)}G^{t,m}, \pi^n LG^{t,m}) + d_F(\pi^n LG^{t,m}, \pi^n LG^t).
\tag{91}
$$

We use lemma 3.5 to control the two terms in the right hand of the inequality respectively.

We first bound $d_F(\pi^n L^{(m)}G^{t,m}, \pi^n LG^{t,m})$. Taking $x = y = G^{t,m}$ and $L' = \frac{1}{n}(G^\star G^{\star\top} + \sigma W^{(m)})$ in lemma 3.5, we verify the conditions in the lemma as follows.

$$
d_F(G^{t,m}, G^\star) \leq d_F(G^{t,m}, G^t) + d_F(G^t, G^\star) \leq (k_1 + k_3)\sqrt{n}.
\tag{92}
$$

Let $d_F(G^{t,m}, G^t) = \|G^{t,m} - G^t Q\|_F$.

$$
\begin{aligned}
\|WG^{t,m}\|_\infty &= \|W(G^{t,m} - G^t Q) + WG^t Q\|_\infty\\
&\leq \|WG^t\|_\infty + \max_{1\leq m\leq n}\|W_m^\top(G^{t,m} - G^t Q)\|_F\\
&\leq \|WG^t\|_\infty + \max_{1\leq m\leq n}\|W_m\|_2 \cdot d_F(G^{t,m}, G^t)\\
&\leq k_2\sqrt{n\log n} + C_2\sqrt{n}k_1\\
&\leq 2k_2\sqrt{n\log n}.
\end{aligned}
\tag{93}
$$

$$
\begin{aligned}
\|W^{(m)}G^{t,m}\|_\infty &\leq \|WG^{t,m}\|_\infty + \|\Delta W^{(m)}G^{t,m}\|_F\\
&\leq k_2\sqrt{n\log n} + C_2 k_1\sqrt{n} + 2C_1\sqrt{r}\sqrt{n\log n}\\
&\leq 2k_2\sqrt{n\log n},
\end{aligned}
\tag{94}
$$

where the second inequality comes from (85). Using lemma 3.5, we have

$$\epsilon = \frac{(k_1 + k_3)^2}{2} + \sigma \cdot 2k_2\sqrt{\log n/n} \le \frac{1}{2}.$$

$$
\begin{aligned}
d_F(\pi^n L^{(m)} G^{t,m}, \pi^n L G^{t,m}) &\le 8 d_F(L^{(m)} G^{t,m}, L G^{t,m}) \\
&\le 8\|(L^{(m)} - L) G^{t,m}\|_F \\
&= 8\|\Delta W^{(m)} G^{t,m}\|_F \cdot \frac{\sigma}{n} \\
&\le \frac{8\sigma}{n} 2C_1\sqrt{r}\sqrt{n\log n} \le \frac{k_1}{2}.
\end{aligned}
\tag{95}
$$

We then bound $d_F(\pi^n L G^{t,m}, \pi^n L G^t)$. Taking $x = G^{t,m}, y = G^t$ and $L' = L$ in lemma 3.5, we verify the conditions in the lemma as follows.

$$d_F(G^{t,m}, G^\star) \le (k_1 + k_3)\sqrt{n}, \quad d_F(G^t, G^\star) \le k_3\sqrt{n},$$
$$\|WG^{t,m}\|_\infty \le 2k_2\sqrt{n\log n}, \quad \|WG^t\|_\infty \le k_2\sqrt{n\log n}.$$

These inequalities are either proved in (a) or coming from the induction assumption. Using lemma 3.5, $\epsilon \le \frac{1}{2}, \rho \le 8(2(k_1 + k_2) + \frac{\sigma\|W\|_2}{n}) \le \frac{1}{2}$, we have

$$d_F(\pi^n L G^{t,m}, \pi^n L G^t) \le \frac{1}{2} d_F(G^{t,m}, G^t) \le \frac{k_1}{2}.
\tag{96}$$

Combining (95), (96), we have:

$$d_F(G^{t+1,m}, G^{t+1}) \le \frac{k_1}{2} + \frac{k_1}{2} = k_1.
\tag{97}$$

**Step 3:** $\|WG^{t+1}\|_\infty \le k_2\sqrt{n\log n}$.

$\|WG^{t+1}\|_\infty = \max_{m\in[n]} \|W_m^\top G^{t+1}\|_F$. Let $d_F(G^{t+1,m}, G^{t+1}) = \|G^{t+1,m} - G^{t+1}Q\|_F$. We derive that $\forall m \in [n]$,

$$
\begin{aligned}
\|W_m^\top G^{t+1}\|_F &\le \|W_m^\top(G^{t+1}Q - G^{t+1,m})\|_F + \|W_m^\top G^{t+1,m}\|_F \\
&\le \|W_m\|_2 d_F(G^{t+1,m}, G^{t+1}) + \|\Delta W^{(m)} G^{t+1,m}\|_F \\
&\le C_2\sqrt{n}k_1 + 2C_1\sqrt{r}\sqrt{n\log n} \\
&\le k_2\sqrt{n\log n}.
\end{aligned}
\tag{98}
$$

Therefore, $\|W_m^\top G^{t+1}\|_F \le k_2\sqrt{n\log n}$. $\qquad\square$

## F. Proof of Theorem 3.8

Since the contractive region is only valid for the initial $3n^2$ points, we split the proof into 2 parts.

**Theorem F.1.** *Under the condition of theorem E.1, with probability at least $1 - O(n^{-2})$, $d_F(G^{t+1}, G^t) \le \frac{1}{2}d_F(G^t, G^{t-1}), \forall t \in \{0, 1, \cdots, 3n^2 - 1\}$, therefore, $d_F(G^{3n^2-1}, G^{3n^2-2}) \le (\frac{1}{2})^{3n^2-3}k_3\sqrt{n} \le \frac{k_3}{4}$*

*Proof.* With probability at least $1 - O(n^{-2})$ the conclusions of theorem E.2 hold. Let $1 \le t \le 3n^2 - 1$.

$$d_F(G^t, G^\star) \le k_3\sqrt{n}, \quad d_F(G^{t-1}, G^\star) \le k_3\sqrt{n},$$
$$\|WG^t\|_\infty \le k_2\sqrt{n\log n}, \quad \|WG^{t-1}\|_\infty \le k_2\sqrt{n\log n}.$$

Using lemma 3.5, $\epsilon = \frac{k_3^2}{2} + \sigma k_2\sqrt{\log n/n} \le \frac{1}{2}, \rho \le 8(2k_3 + \frac{\sigma\|W\|_2}{n}) \le \frac{1}{2}$.

$$d_F(G^{t+1}, G^t) = d_F(\pi^n L G^t, \pi^n L G^{t-1}) \le \frac{1}{2} d_F(G^t, G^{t-1}).
\tag{99}$$

Using (99) repeatedly we have:

$$d_F(G^{3n^2-1}, G^{3n^2-2}) \leq \frac{1}{2^{3n^2-2}} d_F(G^1, G^0)$$

$$\leq \frac{1}{2^{3n^2-2}} (d_F(G^1, G^\star) + d_F(G^0, G^\star))$$

$$\leq \frac{1}{2^{3n^2-2}} \cdot 2k_3\sqrt{n}$$

$$= \frac{1}{2^{3n^2-3}} \cdot k_3\sqrt{n} \leq \frac{k_3}{4}. \tag{100}$$

$\square$

**Proposition F.2.** *Let $A$ be a symmetric real matrix, $x$ be a vector of the same dimension, then*

$$\lambda_{min}(A)\|x\|_2^2 \leq x^\top A x \leq \lambda_{max}(A)\|x\|_2^2, \tag{101}$$

*where $\lambda_{min}(A), \lambda_{max}(A)$ are the smallest and largest eigenvalues of $A$.*

**Theorem F.3.** *Under the condition of theorem E.1, with probability at least $1 - O(n^{-2})$, the following statements hold:*

$$\forall k \geq 0, d_F(G^{T+k}, G^{T+k-1}) \leq 2^{-k} d_F(G^T, G^{T-1}), \tag{102}$$

*where $T = 3n^2 - 1$. In the metric space $(O_{d,r}^n / \sim, d_F)$, $[G^t]$ converge to $[G^\infty]$. $G^\infty \in O_{d,r}^n$ is a fixed point of $\mathcal{P} = \pi^n C$ and satisfies the following properties:*

$$d_F(G^\infty, G^\star) \leq \frac{3}{2} k_3\sqrt{n}, \quad \|WG^\infty\|_\infty \leq (k_2 + C_2 k_3)\sqrt{n \log n}, \tag{103}$$

$$\forall t \geq 0, d_F(G^{t+1}, G^\infty) \leq \frac{d_F(G^t, G^\infty)}{2}, \tag{104}$$

$$\forall i \in [n], G_i^\infty \Lambda_i = (CG^\infty)_i, \quad \Lambda_i = n[(LG^\infty)_i^\top (LG^\infty)_i]^{\frac{1}{2}}. \tag{105}$$

*Proof.* **Step1:** We first prove (102) by induction. When $k = 0$, the statement is trivial. Now we assume that the property holds for all $0 \leq k \leq m, m \geq 0$. We then prove the statement for $k = m + 1$. We first verify the conditions in lemma 3.5 for $x = G^{T+m}, y = G^{T+m-1}$ and $L = L'$.

$$d_F(G^{T+m}, G^\star) \leq d_F(G^{T+m}, G^{T-1}) + d_F(G^{T-1}, G^\star)$$

$$\leq \sum_{l=0}^m d_F(G^{T+l}, G^{T+l-1}) + k_3\sqrt{n}$$

$$\leq (\sum_{l=0}^m 2^{-l}) d_F(G^T, G^{T-1}) + k_3\sqrt{n}$$

$$\leq \frac{k_3}{2} + k_3\sqrt{n} \leq \frac{3k_3}{2}\sqrt{n},$$

where the last inequality follows from $d_F(G^T, G^{T-1}) \leq \frac{k_3}{4}$ in theorem F.1. Similarly, $d_F(G^{T+m-1}, G^\star) \leq \frac{3k_3}{2}\sqrt{n}$.

Assume that $d_F(G^{T+m}, G^{T-1}) = \|G^{T+m} - G^{T-1}Q\|_F$, then

$$\|WG^{T+m}\|_\infty = \|W(G^{T+m} - G^{T-1}Q) + WG^{T-1}Q\|_\infty$$

$$\leq \|W(G^{T+m} - G^{T-1}Q)\|_\infty + \|WG^{T-1}\|_\infty$$

$$\leq k_2\sqrt{n \log n} + \|W(G^{T+m} - G^{T-1}Q)\|_F$$

$$\leq k_2\sqrt{n \log n} + C_2\sqrt{n} d_F(G^{T+m}, G^{T-1}), \tag{106}$$

where the second inequality follows from theorem F.1. By the induction assumption we have

$$d_F(G^{T+m}, G^{T-1}) \leq \sum_{l=0}^{m} d_F(G^{T+l}, G^{T+l-1}) \leq \frac{k_3}{4} \sum_{l=0}^{m} 2^{-l} \leq \frac{k_3}{2}. \tag{107}$$

Combining (106) and (107), we have $\|WG^{T+m}\|_\infty \leq (k_2 + C_2 k_3)\sqrt{n \log n}$. Similarly, we have $\|WG^{T+m-1}\|_\infty \leq (k_2 + C_2 k_3)\sqrt{n \log n}$. Using lemma 3.5, we derive

$$d_F(G^{T+m+1}, G^{T+m}) = d_F(\mathcal{P}G^{T+n}, \mathcal{P}G^{T+n-1}) \leq \frac{1}{2}d_F(G^{T+n}, G^{T+n-1}) \leq \frac{1}{2^{n+1}}d_F(G^T, G^{T-1}), \tag{108}$$

which completes the proof of (102). Therefore $[G^t]$ is an Cauchy sequence in $(O_{d,r}^n / \sim, d_F)$ and must converge to $[G^\infty] \in O_{d,r}^n / \sim$

**Step 2:** Now we proof (103). Since (102) holds for all $k \geq 0$, following the same proof in step 1, we can derive that $\forall m \geq 0$,

$$d_F(G^{T+m}, G^\star) \leq \frac{3k_3}{2}\sqrt{n}, \quad \|WG^{T+m}\|_\infty \leq (k_2 + C_2 k_3)\sqrt{n \log n}. \tag{109}$$

By continuity we derive (103).

**Step 3:** We now prove (104). Combining (109),(103) and using lemma (3.5), we deduce that $\forall t \geq 0, d_F(\mathcal{P}G^t, \mathcal{P}G^\infty) \leq \frac{1}{2}d_F(G^t, G^\infty)$.

$$d_F(\mathcal{P}G^\infty, G^\infty) \leq d_F(\mathcal{P}G^\infty, \mathcal{P}G^t) + d_F(G^{t+1}, G^\infty)$$
$$\leq \frac{1}{2}d_F(G^t, G^\infty) + d_F(G^{t+1}, G^\infty).$$

Letting $t \to \infty$, we get $d_F(PG^\infty, G^\infty) = 0$. Therefore, $\forall t \geq 0$ we have

$$d_F(G^{t+1}, G^\infty) \leq d_F(\mathcal{P}G^t, \mathcal{P}G^\infty) + d_F(\mathcal{P}G^\infty, G^\infty) \leq \frac{1}{2}d_F(G^t, G^\infty). \tag{110}$$

**Step 4:** Lastly, we prove (105). Since $d_F(\mathcal{P}G^\infty, G^\infty) = 0$, there exist a matrix $Q \in O(r)$ such that $\pi^n LG^\infty = G^\infty Q$. $\forall i \in [n], \pi(LG^\infty)_i = G_i^\infty Q$. Assume the SVD decomposition of $\pi(LG^\infty)_i$ is $\pi(LG^\infty)_i = UDV^\top$, where $U \in \mathbb{R}^{d_i \times r}, U^\top U = I_r, D \in \mathbb{R}^{r \times r}, V \in O(r)$.

$$(LG^\infty)_i = UDV^\top = UV^\top V DV^\top$$
$$= (\pi(LG^\infty)_i) \cdot [(LG^\infty)_i^\top (LG^\infty)_i]^{\frac{1}{2}} = G_i^\infty Q \cdot [(LG^\infty)_i^\top (LG^\infty)_i]^{\frac{1}{2}}. \tag{111}$$

Multiplying both sides of (111) with $(G_i^\infty)^\top$ and summing from 1 to $n$, we get

$$(G^\infty)^\top LG^\infty = \sum_{i=1}^{n} (G_i^\infty)^\top (LG^\infty)_i$$
$$= \sum_{i=1}^{n} (G_i^\infty)^\top G_i^\infty Q \cdot [(LG^\infty)_i^\top (LG^\infty)_i]^{\frac{1}{2}} \tag{112}$$
$$= Q \sum_{i=1}^{n} [(LG^\infty)_i^\top (LG^\infty)_i]^{\frac{1}{2}}.$$

Then, we will prove that the matrix $(G^\infty)^\top LG^\infty$ is positive definite. Since $(G^\infty)^\top LG^\infty = \frac{1}{n}(G^\infty)^\top G^\star G^{\star\top} G^\infty + \frac{\sigma}{n}(G^\infty)^\top WG^\infty$. By proposition F.2, $\forall x \in \mathbb{R}^{r \times 1}$ with $\|x\|_2 = 1$,

$$x^\top (G^\infty)^\top LG^\infty x \geq \frac{\sigma}{n}\lambda_{min}((G^\infty)^\top WG^\infty) + \frac{1}{n}\lambda_{min}((G^\infty)^\top G^\star G^{\star\top} G^\infty). \tag{113}$$

We bound the two terms in the right hand of the the inequality respectively.

First, we bound $\lambda_{min}((G^\infty)^\top W G^\infty)$.

$$\lambda_{min}((G^\infty)^\top W G^\infty) \geq -\|(G^\infty)^\top W G^\infty\|_2 \geq -n\sqrt{r}\|W\|_2 \tag{114}$$

Second, we bound $\lambda_{min}((G^\infty)^\top G^\star G^{\star\top} G^\infty)$. Since $d_F(G^\infty, G^\star) \leq \frac{3}{2}k_3\sqrt{n}$, by proposition C.2, we have $(1 - \frac{(\frac{3}{2}k_3)^2}{2})n \leq \sigma_i(G^{\star\top}G^\infty) \leq n$. Therefore

$$\lambda_{min}((G^\infty)^\top G^\star G^{\star\top} G^\infty) = \sigma_r^2(G^{\star\top}G^\infty) \geq (1 - \frac{9}{8}k_3^2)^2 n^2. \tag{115}$$

Substituting (114) and (115) into (113), we have:

$$x^\top (G^\infty)^\top L G^\infty x \geq -\sigma\sqrt{r}\|W\|_2 + (1 - \frac{9}{8}k_3^2)^2 n \geq ((1 - \frac{9}{8}k_3^2)^2 - \frac{1}{12000\sqrt{2}})n > 0, \tag{116}$$

which implies that $(G^\infty)^\top L G^\infty$ is positive definite.

Let $X = (G^\infty)^\top L G^\infty$, $Y = \sum_{i=1}^n [(LG^\infty)_i^\top (LG^\infty)_i]^{\frac{1}{2}}$, then $Y$ is positive semi-definite and $X = QY$. Since $X$ is positive definite, we have $X = X^{\frac{1}{2}}X^{\frac{1}{2}}$. Therefore,

$$X^{-\frac{1}{2}}Q^\top X^{\frac{1}{2}} = X^{-\frac{1}{2}}Y X^{-\frac{1}{2}}. \tag{117}$$

$X^{-\frac{1}{2}}Y X^{-\frac{1}{2}}$ is positive semi-definite and $Q^\top$ is similar to $X^{-\frac{1}{2}}Y X^{-\frac{1}{2}}$, therefore all eigenvalues of $Q$ are positive real numbers. Since $Q \in O(r)$, we have $Q = I_r$, which implies that $G^\infty$ is a fixed point of $\mathcal{P}$. Substituting $Q = I_r$ into (111), we have

$$(CG^\infty)_i = G_i^\infty \cdot n[(LG^\infty)_i^\top (LG^\infty)_i]^{\frac{1}{2}}. \tag{118}$$

$\square$

# G. Proof of Theorem 3.10

**Step 1:** Verify optimality and uniqueness of $G^\infty$ and tightness of SDP.

By lemma 3.9, in order to verify the optimality of $G^\infty$, we only need to prove that $M = L(G^\infty, \Lambda)$ is positive semi-definite and of rank $nd - r$. Since $\forall i \in [n], G_i^\infty \Lambda_i = (CG^\infty)_i, \quad \Lambda_i = n[(LG^\infty)_i^\top (LG^\infty)_i]^{\frac{1}{2}}$, it is easy to show that $MG^\infty = 0$. We can decompose space $R^{nd}$ by $R^{nd} = span(G^\infty) \bigoplus span(G^\infty)^\perp$.

If we can show that $\forall v \in span(G^\infty)^\perp, \|v\|_2 = 1, v^\top M v > 0$, then the proof is complete. The reasons are as follows. On the one hand, since $MG^\infty = 0$ and $\forall v \in span(G^\infty)^\perp, \|v\|_2 = 1, Mv \neq 0$, the solution space of linear equation $Mx = 0$ is exactly $span(G^\infty)$. Therefore, $\text{rank}(M) = nd - dim(span(G^\infty)) = nd - r$. On the other hand, $\forall x \in R^{nd}$, we have decomposition $x = u + v$, where $u \in span(G^\infty), v \in span(G^\infty)^\perp$. Since $MG^\infty = 0, Mu = 0$. Therefore, $x^\top M x = (u^\top + v^\top)M(u + v) = v^\top M v \geq 0$, from which we know that $M$ is positive semi-definite.

Now we show that $\forall v \in span(G^\infty)^\perp, \|v\|_2 = 1, v^\top M v > 0$. We assume that $v^\top = \begin{pmatrix} v_1^\top & \cdots & v_n^\top \end{pmatrix}^\top$ and $v_i \in \mathbb{R}^{d_i \times 1}$.

$$\begin{aligned}
v^\top M v &= \{\sum_{i=1}^n v_i^\top [G_i^\infty \Lambda_i (G_i^\infty)^\top + \tau_i(I_{d_i} - G_i^\infty (G_i^\infty)^\top)]v_i\} - v^\top C v \\
&= \sum_{i=1}^n \tau_i \|v_i\|_2^2 + \sum_{i=1}^n (v_i^\top G_i^\infty \Lambda_i (G_i^\infty)^\top v_i - \tau_i v_i^\top G_i^\infty (G_i^\infty)^\top v_i) - v^\top C v \\
&\geq \sum_{i=1}^n \tau_i \|v_i\|_2^2 - v^\top C v,
\end{aligned} \tag{119}$$

where the inequality comes from proposition F.2, since $v_i^\top G_i^\infty \Lambda_i (G_i^\infty)^\top v_i = ((G_i^\infty)^\top v_i)^\top \Lambda_i (G_i^\infty)^\top v_i \geq \tau_i \|(G_i^\infty)^\top v_i\|_2^2 = \tau_i v_i^\top G_i^\infty (G_i^\infty)^\top v_i$.

First, we provide a upper bound for $v^\top C v$. Let $d_F(G^\infty, G^\star) = \|G^\infty - G^\star Q\|_F$.

$$v^\top C v = v^\top G^\star G^{\star\top} v + \sigma v^\top W v$$

$$\leq \|G^{\star\top} v\|_2^2 + \sigma \lambda_{max}(W) \tag{120}$$

$$\leq \|G^{\star\top} v\|_2^2 + \sigma \|W\|_2,$$

where the second inequality comes from proposition F.2. Since $v \in span(G^\infty)^\perp$,

$$\|G^{\star\top} v\|_2 = \|v^\top (G^\infty - G^\star Q)\|_2 \leq d_F(G^\infty, G^\star). \tag{121}$$

Combining (120) and (121) we have

$$v^\top C v \leq d_F^2(G^\infty, G^\star) + \sigma \|W\|_2 \leq (\frac{3}{2} k_3 \sqrt{n})^2 + \sigma C_2 \sqrt{n} \leq (\frac{9}{4} k_3^2 + \frac{1}{12000\sqrt{2}}) n. \tag{122}$$

Second, we provide a lower bound for $\sum_{i=1}^n \tau_i \|v_i\|_2^2$. Since $\tau_i$ is the smallest eigenvalue of $\Lambda_i = n[(LG^\infty)_i^\top (LG^\infty)_i]^{\frac{1}{2}}$, $\tau_i = n\sigma_r((LG^\infty)_i)$. We only need to bound $\sigma_r((LG^\infty)_i)$. By lemma C.7, we derive

$$\sigma_r((LG^\infty)_i) \geq \frac{1}{n}\sigma_r(G_i^\star G^{\star\top} G^\infty) - \frac{\sigma}{n}\sigma_1((WG^\infty)_i) \geq \frac{1}{n}\sigma_r(G^{\star\top} G^\infty) - \frac{\sigma}{n}\|WG^\infty\|_\infty. \tag{123}$$

Since $d_F(G^\infty, G^\star) \leq \frac{3}{2} k_3 \sqrt{n}$, by proposition C.2, we have $\sigma_r(G^{\star\top} G^\infty) \geq (1 - \frac{9}{8} k_3^2) n$. Therefore,

$$\sigma_r((LG^\infty)_i) \geq (1 - \frac{9}{8} k_3^2) - \frac{\sigma}{n} \cdot 2k_2 \sqrt{n \log n} \geq 1 - \frac{9}{8} k_3^2 - \frac{1}{12000}. \tag{124}$$

Combining (119),(122), (124), we get,

$$v^\top M v \geq \left((1 - \frac{9}{8} k_3^2 - \frac{1}{12000}) - (\frac{9}{4} k_3^2 + \frac{1}{12000\sqrt{2}})\right) n > 0, \tag{125}$$

which completes the proof of the first part of the theorem.

**Step 2:** $d_F(G^\infty, G^\star) \lesssim \sigma$.

Note that $G^\infty$ is a global optimum we have

$$tr(G^{\star\top} C G^\star) = \sum_{i,j=1}^n tr(G_i^{\star\top} C_{ij} G_j^\star) \leq \sum_{i,j=1}^n tr(G_i^\infty C_{ij} G_j^\infty) = tr((G^\infty)^\top C G^\infty). \tag{126}$$

Using the model $C = G^\star G^{\star\top} + \sigma W$, we derive

$$n^2 r + \sigma tr(G^{\star\top} W G^\star) \leq tr\left((G^{\star\top} G^\infty)^\top \cdot G^{\star\top} G^\infty\right) + \sigma tr((G^\infty)^\top W G^\infty). \tag{127}$$

Assuming that the eigenvalues of $G^{\star\top} G^\infty$ are $\sigma_1, \cdots, \sigma_r$, we have

$$\sum_{i=1}^r (n^2 - \sigma_i^2) \leq \sigma \left(tr((G^\infty)^\top W G^\infty) - tr(G^{\star\top} W G^\star)\right). \tag{128}$$

Since $n^2 - \sigma_i^2 = (n - \sigma_i)(n + \sigma_i) \geq n(n - \sigma_i)$, we have

$$\sum_{i=1}^r (n - \sigma_i) \leq \frac{\sigma}{n} \left(tr((G^\infty)^\top W G^\infty) - tr(G^{\star\top} W G^\star)\right). \tag{129}$$

Assume that $d_F(G^\star, G^\infty) = \|G^\infty - G^\star Q\|_F$. The difference of quadratic terms can be bounded by

$$tr((G^\infty)^\top W G^\infty) - tr(G^{\star\top} W G^\star) = tr\left((G^\infty - G^\star Q)^\top W (G^\infty + G^\star Q)\right)$$

$$\leq \|(G^\infty - G^\star Q)^\top\|_F \cdot \|W(G^\infty + G^\star Q)\|_F \tag{130}$$

$$\leq d_F(G^\star, G^\infty) \cdot C_2 \sqrt{n} \cdot 2\sqrt{nr} = 2C_2 \sqrt{rn} d_F(G^\star, G^\infty).$$

Using the fact that $d_F^2(G^\star, G^\infty) = 2(nr - \|G^{\star\top}G^\infty\|_\star)$ and combining (129) and (130), we derive

$$\frac{1}{2}d_F^2(G^\star, G^\infty) \leq \frac{\sigma}{n} \cdot 2C_2\sqrt{r}nd_F(G^\star, G^\infty). \tag{131}$$

Therefore, we get

$$d_F(G^\star, G^\infty) \leq 4C_2\sqrt{r}\sigma. \tag{132}$$

**Step 3:** $d_\infty(G^\infty, G^\star) \lesssim \sigma\sqrt{\frac{\log n}{n}}$

Since $G^\infty$ is a fixed point of $\pi^n L = \pi^n\frac{C}{n}$, $G_i^\infty = \pi((LG^\infty)_i)$. Let $d_F(G^\infty, G^\star) = \|G^\infty - G^\star Q\|_F$. Using lemma C.5 we derive

$$\|G_i^\infty - G_i^\star Q\|_F = \|\pi((LG^\infty)_i) - \pi(G_i^\star Q)\|_F \leq 4\|(LG^\infty)_i - G_i^\star Q\|_F. \tag{133}$$

Noticing that $(LG^\infty)_i = \frac{1}{n}G_i^\star G^{\star\top}G^\infty + \frac{\sigma}{n}(WG^\infty)_i$, we have

$$\begin{aligned}
\|(LG^\infty)_i - G_i^\star Q\|_F &= \|\frac{1}{n}G_i^\star(G^{\star\top}G^\infty - nQ) + \frac{\sigma}{n}(WG^\infty)_i\|_F \\
&\leq \frac{1}{n}\|G^{\star\top}G^\infty - nQ\|_F + \frac{\sigma}{n}\|WG^\infty\|_\infty \\
&\leq \frac{1}{n}\|G^{\star\top}G^\infty - nQ\|_F + \frac{\sigma}{n}(k_2 + C_2k_3)\sqrt{n\log n}.
\end{aligned} \tag{134}$$

Now we bound $\|G^{\star\top}G^\infty - nQ\|_F$.

$$\begin{aligned}
\|G^{\star\top}G^\infty - nQ\|_F^2 &= \|G^{\star\top}G^\infty\|_F^2 + n^2r - 2n\langle Q, G^{\star\top}G^\infty\rangle \\
&= \sum_{i=1}^r \sigma_i^2(G^{\star\top}G^\infty) + n^2r - 2n\|G^{\star\top}G^\infty\|_\star \\
&= \sum_{i=1}^r (n - \sigma_i(G^{\star\top}G^\infty))^2 \\
&\leq (n - \sigma_r(G^{\star\top}G^\infty))\sum_{i=1}^r n - \sigma_i(G^{\star\top}G^\infty) \\
&\leq \frac{9}{8}k_3^2n(nr - \|G^{\star\top}G^\infty\|_\star) \\
&= \frac{9}{16}k_3^2nd_F^2(G^\star, G^\infty),
\end{aligned} \tag{135}$$

where the second equality is because $Q = argmin_{Q\in O(r)}\|G^\infty - G^\star Q\|_F = argmax_{Q\in O(r)}\langle Q, G^{\star\top}G^\infty\rangle$. The second inequality comes from $d_F(G^\star, G^\infty) \leq \frac{3}{2}k_3\sqrt{n}$ and proposition C.2. Combining (134) and (135) we derive

$$\|(LG^\infty)_i - G_i^\star Q\|_F \leq \frac{3}{4\sqrt{n}}k_3d_F(G^\star, G^\infty) + \frac{\sigma}{n}(k_2 + C_2k_3)\sqrt{n\log n} \leq \frac{3}{4\sqrt{n}}k_3 \cdot 4\sqrt{r}C_2\sigma + (k_2 + C_2k_3)\sigma\sqrt{\log n/n}. \tag{136}$$

Substitute the above equation into (133) we derive

$$d_\infty(G^\infty, G^\star) \leq \|G^\infty - G^\star Q\|_\infty \leq 4(\frac{3}{4\sqrt{n}}k_3 \cdot 4\sqrt{r}C_2\sigma + (k_2 + C_2k_3)\sigma\sqrt{\log n/n}) \lesssim \sigma\sqrt{\log n/n}. \tag{137}$$

# H. Proof of Lemma 3.12

**Lemma H.1.** *If $p > \frac{c_0\log n}{n}$, then with probability at least $1 - O(n^{-2})$, for all $m \in [n]$,*

$$\|W_\star\|_2 \leq C_1\sqrt{np}, \quad \|W_\star^{(m)}\|_2 \leq C_1\sqrt{np}, \quad \|\Delta W_\star^{(m)}\|_2 \leq C_1\sqrt{np}, \quad \|(W_\star)_m\|_2 \leq C_1\sqrt{np},$$

$$\|\Delta_\star\|_2 \leq C_1\sqrt{np}, \quad \|\Delta_\star^{(m)}\|_2 \leq C_1\sqrt{np}, \quad \|\Delta(\Delta_\star)^{(m)}\|_2 \leq C_1\sqrt{np}, \quad \|(\Delta_\star)_m\|_2 \leq C_1\sqrt{np},$$

*where the notation follows that of Lemma A.1.*

**Lemma H.2.** *Let $X_1, X_2, \cdots X_N$ be independent mean zero random variables such that $\forall i, |X_i| \leq K$, then for any $t \geq 0$ we have*

$$P(|\sum_{i=1}^N X_i| \geq t) \leq 2exp\left(-\frac{t^2/2}{\sum_{i=1}^N \mathbb{E}X_i^2 + Kt/3}\right) \tag{138}$$

*Proof.* This is theorem 2.9.5 in (Vershynin, 2018) □

**Lemma H.3.** *Let $X_1, X_2, \cdots X_N$ be independent mean zero sub-gaussian random variables. Then for any $t \geq 0$ we have*

$$P(|\sum_{i=1}^N X_i| \geq t) \leq 2exp\left(-\frac{ct^2}{\sum_{i=1}^N \|X_i\|_{\psi_2}^2}\right) \tag{139}$$

*Proof.* This is theorem 2.7.3 in (Vershynin, 2018) □

**Lemma H.4.** *let $U^{(m)}$, $1 \leq m \leq n$ be sets whose elements are random matrices. $\forall m \in [n], \forall u \in U^{(m)}$, we have $u \in O_{d,r}^n$, and $u$ is independent from $\Delta W_\star^{(m)}$. Moreover, $\forall 1 \leq m \leq n$, cardinality of $U^{(m)}$ is not random and is not larger than $3n^2$. $p > \frac{(\log n)^2}{n}$, $n \geq 2$. Then with probability at least $1 - O(n^{-2})$, the following holds:*

$$\forall 1 \leq m \leq n, \quad \max_{u \in U^{(m)}} \|\Delta W_\star^{(m)}u\|_F \leq C_2\sqrt{np\log n}, \tag{140}$$

*where $C_2$ is an absolute constant.*

*Proof.* Similar to lemma 3.4 We only prove that for a fixed $m$ and a non-random $U^{(m)}$, with probability at least $1 - O(n^{-3})$, $\max_{u \in U^{(m)}} \|\Delta W^{(m)}u\|_F \leq C_1\sqrt{n\log n}$.

Since all entries of $W$ are sub-gaussian random variables, there exists an absolute constant $c$, such that with probability at least $1 - O(n^{-5})$, all elements of $W$ have absolute values smaller than $M = c\sqrt{\log n}$. In another word, event $B = \{$All elements of $W$ have absolute values smaller than $M\}$ happens with probability at least $1 - O(n^{-5})$. In order to use lemma H.2, we define the truncated noise matrix $\bar{W}$ by $\bar{W}_{mk}^{ij} := W_{mk}^{ij} \cdot \mathbb{I}_{\{|W_{mk}^{ij}| \leq M\}}$, where $W_{mk}^{ij}$ denotes the $(i,j)$ element of the $(m,k)$ block of $W$ and $\mathbb{I}$ denotes the indicator function.

Letting $u \in U^{(m)}$, we have

$$\|\Delta W_\star^{(m)}u\|_F^2 = \|(\Delta W_\star^{(m)}u)_m\|_F^2 + \sum_{k \neq m} \|(\Delta W_\star^{(m)}u)_k\|_F^2. \tag{141}$$

We control the first term and the second on the right hand of the equation respectively.

**Step 1:** For the first term $(\Delta W_\star^{(m)}u)_m = \sum_{k=1}^n (W_\star)_{mk}u_k$, we have

$$(\Delta W_\star^{(m)}u)_m^{ij} = \sum_{k=1}^n e_{mk} \sum_{s=1}^{d_k} W_{mk}^{is}u_k^{sj}. \tag{142}$$

Using law of total probability, $\forall t \geq 0$,

$$P\left(\left|(\Delta W_\star^{(m)}u)_m^{ij}\right| \geq t\right) = P\left(\left|(\Delta W_\star^{(m)}u)_m^{ij}\right| \geq t, B\right) + P\left(\left|(\Delta W_\star^{(m)}u)_m^{ij}\right| \geq t, B^c\right)$$

$$\leq P\left(\left|\sum_{k=1}^n e_{mk} \sum_{s=1}^{d_k} \bar{W}_{mk}^{is}u_k^{sj}\right| \geq t\right) + P(B^c) \tag{143}$$

$$\leq P\left(\left|\sum_{k=1}^n e_{mk} \sum_{s=1}^{d_k} \bar{W}_{mk}^{is}u_k^{sj}\right| \geq t\right) + O(n^{-5}).$$

Since $W_{mk}^{is}$ are independent and have symmetric distribution with respect to $y$-axis, $\bar{W}_{mk}^{is}$ are also independent and have symmetric distribution with respect to $y$-axis. Therefore, for fixed $(i,j)$, $\{e_{mk}\sum_{s=1}^{d_k}\bar{W}_{mk}^{is}u_k^{sj}\}_{k\in[n]}$ are $n$ mean zero independent random variables.

$$\sum_{k=1}^{n}\mathbb{E}\left((e_{mk}\sum_{s=1}^{d_k}\bar{W}_{mk}^{is}u_k^{sj})^2\right) = p\sum_{k=1}^{n}\sum_{s=1}^{d_k}(u_k^{sj})^2\mathbb{E}(\bar{W}_{mk}^{is})^2 \leq arnp, \tag{144}$$

where $\mathbb{E}(W_{mk}^{is})^2 = a$. Noticing that $|u_k^{sj}| \leq \|u_k\|_F = \sqrt{r}$, we have $|e_{mk}\sum_{s=1}^{d_k}\bar{W}_{mk}^{is}u_k^{sj}| \leq c\sqrt{r}\bar{d}\sqrt{\log n}$, where $\bar{d} = \max_{k\in[n]}d_k$. Using lemma H.2,

$$P(|\sum_{k=1}^{n}e_{mk}\sum_{s=1}^{d_k}\bar{W}_{mk}^{is}u_k^{sj}| \geq t) \leq 2exp\left(-\frac{t^2/2}{arnp + c\sqrt{r}\bar{d}\sqrt{\log n}\frac{t}{3}}\right). \tag{145}$$

Take $t = c_1\sqrt{np\log n}$. Since $p > \frac{(\log n)^2}{n}$, with probability at least $1 - O(n^{-5})$, $|(\Delta W_\star^{(m)}u)_m^{ij}| \leq c_1\sqrt{np\log n}$. Therefore, $\|(\Delta W_\star^{(m)}u)_m\|_F \lesssim \sqrt{np\log n}$.

**Step 2:** For the second term,

$$\sum_{k\neq m}\|(\Delta W_\star^{(m)}u)_k\|_F^2 = \sum_{k\neq m}\|(W_\star)_{km}u_m\|_F^2 \leq \|u_m\|_2^2\sum_{k\neq m}e_{km}\|W_{km}\|_F^2 = \sum_{k\neq m}e_{km}\|W_{km}\|_F^2. \tag{146}$$

Let $\mu_{km} = \mathbb{E}(e_{km}\|\bar{W}_{km}\|_F^2)$. It is obvious that $\mu_{km} \leq \mathbb{E}(e_{km}\|W_{km}\|_F^2) = pd_kd_ma \leq a\bar{d}^2p$. Besides, $\{e_{km}\|\bar{W}_{km}\|_F^2 - \mu_{km}\}_{k\neq m}$ are mean zero, independent random variables. Let $\mathbb{E}\left((W_{mk}^{ij})^4\right) = b$, then

$$\mathbb{E}\left(e_{km}\|\bar{W}_{km}\|_F^2 - \mu_{km}\right)^2 \leq \mathbb{E}(e_{km}\|\bar{W}_{km}\|_F^2)^2 \leq \mathbb{E}(e_{km}\|W_{km}\|_F^2)^2 = \left(b + (d_kd_m - 1)a^2\right)d_kd_mp. \tag{147}$$

Therefore, $\sum_{k\neq m}\mathbb{E}\left(e_{km}\|\bar{W}_{km}\|_F^2 - \mu_{km}\right)^2 \leq c_2np$.

Since $|\bar{W}_{km}^{ij}| \leq c\sqrt{\log n}$, we have

$$|e_{mk}\|\bar{W}_{km}\|_F^2 - \mu_{km}| \leq c^2d_md_k\log n + pd_kd_ma \leq c_3\log n. \tag{148}$$

Using lemma H.2, we have

$$P\left(\left|\sum_{k\neq m}\left(e_{km}\|\bar{W}_{km}\|_F^2 - \mu_{km}\right)\right| \geq t\right) \leq 2exp\left(-\frac{t^2/2}{c_2np + c_3\log n\frac{t}{3}}\right). \tag{149}$$

Since $p \geq \frac{(\log n)^2}{n}$, take $t = c_4np$, then with probability at least $1 - O(n^{-5})$, $\left|\sum_{k\neq m}\left(e_{km}\|\bar{W}_{km}\|_F^2 - \mu_{km}\right)\right| \leq c_4np$. Therefore, $\sum_{k\neq m}e_{km}\|\bar{W}_{km}\|_F^2 \leq c_4np + \sum_{k\neq m}\mu_{km} \leq c_5np$. Similar to step 1, we have decomposition

$$P\left(\sum_{k\neq m}e_{km}\|W_{km}\|_F^2 \geq c_5np\right) = P\left(\sum_{k\neq m}e_{km}\|W_{km}\|_F^2 \geq c_5np, B\right) + P\left(\sum_{k\neq m}e_{km}\|W_{km}\|_F^2 \geq c_5np, B^c\right)$$

$$\leq P\left(\sum_{k\neq m}e_{km}\|\bar{W}_{km}\|_F^2 \geq c_5np\right) + O(n^{-5}) \lesssim n^{-5}. \tag{150}$$

Therefore, with probability at least $1 - O(n^{-5})$, $\sum_{k\neq m}\|(\Delta W_\star^{(m)}u)_k\|_F^2 \lesssim np$.

Combing step 1 and step 2 and taking a union bound over $U^{(m)}$, we get the desired result. $\qquad\square$

**Lemma H.5.** *let $U^{(m)}$, $1 \leq m \leq n$ be sets whose elements are random matrices. $\forall m \in [n], \forall u \in U^{(m)}$ we have $u \in O_{d,r}^n$, and $u$ is independent from $\Delta(\Delta_\star)^{(m)}$. Moreover, $\forall 1 \leq m \leq n$, cardinality of $U^{(m)}$ is not random and is not larger than $3n^2$. $p > \frac{\log n}{n}, n \geq 2$. Then with probability at least $1 - O(n^{-2})$, the following holds:*

$$\forall 1 \leq m \leq n, \quad \max_{u\in U^{(m)}}\|\Delta(\Delta_\star)^{(m)}u\|_F \leq C_2\sqrt{np\log n}, \tag{151}$$

*where $C_2$ is an absolute constant.*

*Proof.* Similar to lemma H.4, we have

$$\|\Delta(\Delta_\star)^{(m)}u\|_F^2 = \|(\Delta(\Delta_\star)^{(m)}u)_m\|_F^2 + \sum_{k \neq m} \|(\Delta(\Delta_\star)^{(m)}u)_k\|_F^2. \tag{152}$$

**Step 1:** $\Delta(\Delta_\star)^{(m)}u)_m = \sum_{k=1}^n (e_{mk} - p)G_m^\star G_k^{\star\top} u_k.$

$$\Delta(\Delta_\star)^{(m)}u)_m^{ij} = \sum_{k=1}^n \left( (e_{mk} - p) \sum_{s=1}^{d_k} (G_m^\star G_k^{\star\top})^{is} u_k^{sj} \right). \tag{153}$$

$\{(e_{mk} - p) \sum_{s=1}^{d_k} (G_m^\star G_k^{\star\top})^{is} u_k^{sj}\}_{k \in [n]}$ are mean zero independent random variables. We will use lemma H.2 on them.

$$\left| (e_{mk} - p) \sum_{s=1}^{d_k} (G_m^\star G_k^{\star\top})^{is} u_k^{sj} \right| \leq \sum_{s=1}^{d_k} \left| (G_m^\star G_k^{\star\top})^{is} \right| \cdot |u_k^{sj}| \leq \sum_{s=1}^{d_k} \|G_m^\star G_k^{\star\top}\|_F \|u_k\|_F \leq r\bar{d}. \tag{154}$$

$$\begin{aligned}
\sum_{k=1}^n \mathbb{E}\left( (e_{mk} - p) \sum_{s=1}^{d_k} (G_m^\star G_k^{\star\top})^{is} u_k^{sj} \right)^2 &= p(1-p) \sum_{k=1}^n \left( \sum_{s=1}^{d_k} (G_m^\star G_k^{\star\top})^{is} u_k^{sj} \right)^2 \\
&\leq p \sum_{k=1}^n \left( \sum_{s=1}^{d_k} \left( (G_m^\star G_k^{\star\top})^{is} \right)^2 \right) \cdot \left( \sum_{s=1}^{d_k} \left( u_k^{sj} \right)^2 \right) \\
&\leq p \sum_{k=1}^n \|G_m^\star G_k^{\star\top}\|_F^2 \cdot \left( \sum_{s=1}^{d_k} \left( u_k^{sj} \right)^2 \right) \\
&= pr \sum_{k=1}^n \sum_{s=1}^{d_k} (u_k^{sj})^2 \leq pr\|u\|_F^2 = r^2 np,
\end{aligned} \tag{155}$$

where the second inequality comes from Cauchy inequality. Using lemma H.2, we derive

$$P\left( \left| \sum_{k=1}^n \left( (e_{mk} - p) \sum_{s=1}^{d_k} (G_m^\star G_k^{\star\top})^{is} u_k^{sj} \right) \right| \geq t \right) \leq 2exp\left( -\frac{t^2/2}{r^2 np + r\bar{d}\frac{t}{3}} \right). \tag{156}$$

Taking $t = c_1\sqrt{np\log n}$, we have that with probability at least $1 - O(n^{-5})$, $|\Delta(\Delta_\star)^{(m)}u)_m^{ij}| \lesssim \sqrt{np\log n}$. Therefore, $(\Delta(\Delta_\star)^{(m)}u)_m\|_F \lesssim \sqrt{np\log n}$.

**Step 2:** For the second term,

$$\begin{aligned}
\sum_{k \neq m} \|(\Delta(\Delta_\star)^{(m)}u)_k\|_F^2 &= \sum_{k \neq m} \|(\Delta_\star)_{km}u_m\|_F^2 \leq \sum_{k \neq m} \|(\Delta_\star)_{km}\|_F^2 \\
&= \sum_{k \neq m} (e_{km} - p)^2 \|G_k^\star G_m^{\star\top}\|_F^2 \leq r \sum_{k=1}^n (e_{mk} - p)^2.
\end{aligned} \tag{157}$$

Using lemma H.2 for $\{(e_{mk} - p)^2 - p(1-p)\}_{k \in [n]}$, we can derive that with probability at least $1 - O(n^{-5})$, $\sum_{k \neq m} \|(\Delta(\Delta_\star)^{(m)}u)_k\|_F^2 \lesssim np$.

Combining step 1 and step 2, we get the desired result. $\qquad\square$

**Lemma H.6.** *Let* $C_p^{(m)} = G^\star G^{\star\top} + \widetilde{W}^{(m)} = G^\star G^{\star\top} + \frac{1}{p}\Delta_\star^{(m)} + \frac{\sigma}{p}W_\star^{(m)}$, $\widetilde{G}^{(m)}$ *be the top $r$ leading eigenvectors of* $C_p^{(m)}$ *with* $(\widetilde{G}^{(m)})^\top \widetilde{G}^{(m)} = nI_r$. $\sigma \lesssim \sqrt{np/\log n}$, $p > c\frac{\log n}{\sqrt{n}}$, *Then with probability at least $1 - O(n^{-3})$, $\forall m \in [n]$ we have*

$$\|\Delta W_\star^{(m)} \widetilde{G}^{(m)}\|_F \leq C_3\sqrt{n\log n}. \tag{158}$$

*Proof.* We only need to modify the proof of lemma H.4. Let $u = \widetilde{G}^{(m)}$.

$$\|\Delta W_\star^{(m)} u\|_F^2 = \|(\Delta W_\star^{(m)} u)_m\|_F^2 + \sum_{k \neq m} \|(\Delta W_\star^{(m)} u)_k\|_F^2. \tag{159}$$

**Step 1:** For the first term, $\{e_{mk} \sum_{s=1}^{d_k} W_{mk}^{is} u_k^{sj}\}_{k \in [n]}$ are $n$ mean zero independent sub-Gaussian random variables.

$$\|e_{mk} \sum_{s=1}^{d_k} W_{mk}^{is} u_k^{sj}\|_{\psi_2}^2 \leq \|\sum_{s=1}^{d_k} W_{mk}^{is} u_k^{sj}\|_{\psi_2}^2 \leq c \sum_{s=1}^{d_k} \|W_{mk}^{is} u_k^{sj}\|_{\psi_2}^2 = c \sum_{s=1}^{d_k} (u_k^{sj})^2. \tag{160}$$

Using Hoeffding inequality we have

$$P\left(\left|\sum_{k=1}^{n} \left(e_{mk} \sum_{i=1}^{d_k} W_{mk}^{is} u_k^{sj}\right)\right| \geq t\right) \leq 2exp\left(-\frac{ct^2}{\sum_{k=1}^{n} \sum_{s=1}^{d_k} (u_k^{sj})^2}\right) \leq 2exp(-\frac{ct^2}{nr}). \tag{161}$$

Letting $t = c_1 \sqrt{n \log n}$, we have that with probability at least $1 - O(n^{-4})$, $\|(\Delta W_\star^{(m)} u)_m\|_F \lesssim \sqrt{n \log n}$.

**Step 2:** For the second term, with probability at least $1 - O(n^{-4})$,

$$\sum_{k \neq m} \|(\Delta W_\star^{(m)} u)_k\|_F^2 = \sum_{k \neq m} \|(W_\star)_{km} u_m\|_F^2 \leq \|u_m\|_2^2 \sum_{k \neq m} e_{mk} \|W_{km}\|_F^2$$

$$\leq \frac{64r}{49} \sum_{k \neq m} e_{mk} \|W_{km}\|_F^2 \leq \frac{64r}{49} \sum_{k \neq m} \|W_{km}\|_F^2 \leq cn, \tag{162}$$

where the second to last inequality comes from (167) and the last inequality comes from (27). Combing step 1 and step 2 and taking a union bound over $m \in [n]$, we get the desired result. $\qquad\square$

**Lemma H.7.** *Let $\widetilde{G}^{(m)}$ be the top $r$ leading eigenvectors of $C_p^{(m)}$ with $(\widetilde{G}^{(m)})^\top \widetilde{G}^{(m)} = nI_r$. $\sigma \lesssim \sqrt{np/\log n}$, $p > c\frac{\log n}{\sqrt{n}}$. Then with probability at least $1 - O(n^{-3})$, $\|\widetilde{G}_m^{(m)}\|_F \leq \frac{8}{7}\sqrt{r}$ and $\forall i \in [n] \setminus \{m\}$, $\|\widetilde{G}_i^{(m)}\|_F \leq C_4 \frac{n^{\frac{1}{4}}}{\sqrt{\log n}}$.*

*Proof.* Since $\widetilde{G}^{(m)}$ is the eigenvectors of $C_p^{(m)}$, we have

$$\widetilde{G}^{(m)} \Lambda = C_p^{(m)} \widetilde{G}^{(m)} = G^\star G^{\star\top} \widetilde{G}^{(m)} + \widetilde{W}^{(m)} \widetilde{G}^{(m)}, \tag{163}$$

where $\Lambda \in \mathbb{R}^{r \times r}$ is a diagonal matrix of the top $r$ eigenvalues of $C_p^{(m)}$. Using wely inequality, we have

$$|\lambda_r(C_p^{(m)}) - n| \leq \|\widetilde{W}^{(m)}\|_2, \quad |\lambda_{r+1}(C_p^{(m)}) - 0| \leq \|\widetilde{W}^{(m)}\|_2. \tag{164}$$

Since $\sigma \lesssim \sqrt{np/\log n}$, $p > c\frac{\log n}{n}$,

$$|\widetilde{W}^{(m)}\|_2 \leq \frac{1}{p}\|\Delta_\star^{(m)}\|_2 + \sigma\frac{1}{p}\|W_\star^{(m)}\|_2 \leq C_1(\sqrt{n/p} + \sigma\sqrt{n/p}) \leq \frac{1}{8}n, \tag{165}$$

where the second inequality comes from lemma H.1. Therefore, $\Lambda$ is invertible and $\lambda_r(C_p^{(m)}) \geq \frac{7}{8}n$, $\lambda_{r+1}(C_p^{(m)}) \leq \frac{1}{8}n$. Using (163), we derive that $\forall i \in [n]$

$$\widetilde{G}_i^{(m)} = G_i^\star G^{\star\top} \widetilde{G}^{(m)} \Lambda^{-1} + (\widetilde{W}^{(m)} \widetilde{G}^{(m)})_i \Lambda^{-1}. \tag{166}$$

When $i = m$,

$$\|\widetilde{G}_m^{(m)}\|_F = \|G_i^\star G^{\star\top} \widetilde{G}^{(m)} \Lambda^{-1}\|_F \leq \|G^{\star\top}\|_2 \cdot \|\widetilde{G}^{(m)}\|_F \frac{1}{\lambda_r(C_p^{(m)})} \leq \frac{8}{7}\sqrt{r}. \tag{167}$$

When $i \neq m$,

$$\|\widetilde{G}_i^{(m)}\|_F \leq \frac{8}{7}\sqrt{r} + \frac{8}{7n}\|(\widetilde{W}^{(m)} \widetilde{G}^{(m)})_i\|_F. \tag{168}$$

Next, we will provide a bound for $\|(\widetilde{W}^{(m)}\widetilde{G}^{(m)})_i\|_F$.

$$
\begin{aligned}
\|(\widetilde{W}^{(m)}\widetilde{G}^{(m)})_i\|_F &= \|\frac{1}{p}(\Delta_\star^{(m)}\widetilde{G}^{(m)})_i + \sigma\frac{1}{p}(W_\star^{(m)}\widetilde{G}^{(m)})_i\|_F \\
&\leq \frac{1}{p}\|\Delta_\star^{(m)}\widetilde{G}^{(m)}\|_F + \frac{\sigma}{p}\|((W_\star^{(m)})_i^\top\widetilde{G}^{(m)}\|_F.
\end{aligned}
\tag{169}
$$

For the first term,

$$
\|\Delta_\star^{(m)}\widetilde{G}^{(m)}\|_F \leq \|\Delta_\star^{(m)}\|_2\|\widetilde{G}^{(m)}\|_F \leq C_1 n\sqrt{pr}.
\tag{170}
$$

For the second term,

$$
\|((W_\star^{(m)})_i^\top\widetilde{G}^{(m)}\|_F \leq \|(W_\star)_i^\top\widetilde{G}^{(m)}\|_F + \|(W_\star)_{mi}^\top\widetilde{G}_m^{(m)}\|_F.
\tag{171}
$$

This inequality is because the only difference between $(W_\star^{(m)})_i$ and $(W_\star)_i$ is the $(W_\star)_{mi}$ block. Assuming that $d_F(\widetilde{G}^{(m)}, \widetilde{G}^{(i)}) = \|\widetilde{G}^{(m)} - \widetilde{G}^{(i)}Q)\|_F$, we have

$$
\begin{aligned}
\|(W_\star)_i^\top\widetilde{G}^{(m)}\|_F &\leq \|(W_\star)_i^\top\widetilde{G}^{(i)}\|_F + \|(W_\star)_i^\top(\widetilde{G}^{(m)} - \widetilde{G}^{(i)}Q)\|_F \\
&\leq \|(W_\star)_i^\top\widetilde{G}^{(i)}\|_F + C_1\sqrt{np}d_F(\widetilde{G}^{(m)}, \widetilde{G}^{(i)}) \\
&\leq \|\Delta W_\star^{(i)}\widetilde{G}^{(i)}\|_F + C_1\sqrt{np}\left(d_F(\widetilde{G}^{(m)}, \widetilde{G}) + d_F(\widetilde{G}^{(i)}, \widetilde{G})\right),
\end{aligned}
\tag{172}
$$

where $\widetilde{G}$ and $\widetilde{G}^{(i)}$ are the top $r$ eigenvectors of $C_p$ and $C_p^{(i)}$ respectively, with $\widetilde{G}^\top\widetilde{G} = nI_r = (\widetilde{G}^{(i)})^\top\widetilde{G}^{(i)}$. Using Davis-Kahn inequality, we have

$$
\begin{aligned}
d_F(\widetilde{G}^{(m)}, \widetilde{G}) &\leq \frac{\sqrt{2}\|\Delta\widetilde{W}^{(m)}\widetilde{G}^{(m)}\|_F}{\delta(C_P^{(m)}) - \|\Delta\widetilde{W}^{(m)}\|_2} \leq \frac{8\sqrt{2}}{5n}\|\Delta\widetilde{W}^{(m)}\widetilde{G}^{(m)}\|_F \\
&\leq \frac{8\sqrt{2}}{5n}\|(\frac{1}{p}\Delta(\Delta_\star)^{(m)}\widetilde{G}^{(m)} + \frac{\sigma}{p}\Delta W_\star^{(m)}\widetilde{G}^{(m)}\|_F \\
&\leq \frac{8\sqrt{2}}{5n}(n\sqrt{r/p} + \frac{\sigma}{p}\|\Delta W_\star^{(m)}\widetilde{G}^{(m)}\|_F).
\end{aligned}
\tag{173}
$$

Similarly, we have

$$
d_F(\widetilde{G}^{(i)}, \widetilde{G}) \leq \frac{8\sqrt{2}}{5n}(n\sqrt{r/p} + \frac{\sigma}{p}\|\Delta W_\star^{(i)}\widetilde{G}^{(i)}\|_F).
\tag{174}
$$

Using lemma H.6, we have $\forall i \in [n], \|\Delta W_\star^{(i)}\widetilde{G}^{(i)}\|_F \leq C_3\sqrt{n\log n}$. Substituting the inequality into (172), (173), (174), we have

$$
d_F(\widetilde{G}^{(m)}, \widetilde{G}) \lesssim \frac{1}{\sqrt{p}}, \quad d_F(\widetilde{G}^{(i)}, \widetilde{G}) \lesssim \frac{1}{\sqrt{p}}, \quad \|(W_\star)_i^\top\widetilde{G}^{(m)}\|_F \lesssim \sqrt{n\log n}.
\tag{175}
$$

For the term $\|(W_\star)_{mi}^\top\widetilde{G}_m^{(m)}\|_F$. Since $(W_\star)_{mi} = e_{mi}W_{mi}$ and $W_{mi}$ is a $d_m \times d_i$ matrix with independent mean zero sub-Gaussian entries, it is easy to show that $\|(W_\star)_{mi}\|_2 \lesssim \sqrt{\log n}$ through the standard random matrix norm bound. Therefore, we have

$$
\|(W_\star)_{mi}^\top\widetilde{G}_m^{(m)}\|_F \leq \|(W_\star)_{mi}\|_2 \cdot \|\widetilde{G}_m^{(m)}\|_F \lesssim \sqrt{n\log n}.
\tag{176}
$$

Combing (175),(176) and (171), we have $\|((W_\star^{(m)})_i^\top\widetilde{G}^{(m)}\|_F \lesssim \sqrt{n\log n}$. Substituting this inequality and (170) into (169) we derive $\|(\widetilde{W}^{(m)}\widetilde{G}^{(m)})_i\|_F \lesssim \frac{n}{\sqrt{p}}$. Substituting the inequality into (168), we have $\|\widetilde{G}_i^{(m)}\|_F \lesssim \frac{1}{\sqrt{p}} \lesssim \frac{n^{\frac{1}{4}}}{\sqrt{\log n}}$, Where the last inequality is from the condition $p > c\frac{\log n}{\sqrt{n}}$. $\qquad\square$

**Lemma H.8.** *Let $\sigma \lesssim \sqrt{np/\log n}$, $p > c\frac{\log n}{\sqrt{n}}$. Then with probability at least $1 - O(n^{-2})$, $\forall m \in [n]$, we have*

$$
\|\Delta W_\star^{(m)}\widetilde{G}^{(m)}\|_F \leq C_5\sqrt{np\log n}
\tag{177}
$$

*Proof.* Letting $u = \widetilde{G}^{(m)}$, we have

$$\|\Delta W_\star^{(m)} u\|_F^2 = \|(\Delta W_\star^{(m)} u)_m\|_F^2 + \sum_{k \neq m} \|(\Delta W_\star^{(m)} u)_k\|_F^2. \tag{178}$$

**Step 1:** For the first term, we only need to modify the proof of lemma H.4 by replacing the previous bound $|u_k^{sj}| \leq \sqrt{r}$ by the bound $|u_k^{sj}| \leq C_4 \frac{n^{\frac{1}{4}}}{\sqrt{\log n}}$ in lemma H.7. Similar to the proof of lemma H.4, we define the truncated noise matrix $\bar{W}$. Besides, we define $\bar{u}$ by $\bar{u}_m^{ij} = u_m^{ij} \cdot \mathbb{I}_{\{|u_m^{ij}| \leq M'\}}$, where $M' = C_4 \frac{n^{\frac{1}{4}}}{\sqrt{\log n}}$, $\mathbb{I}$ is the indicator function and $u_m^{ij}$ is the $(i, j)$ element of the $m$ block of $u$. Define event $A = \{$All elements of $u$ have absolute values smaller than $M'\}$. By lemma H.7, $P(A) \geq 1 - O(n^{-3})$. Using (143), we have

$$
\begin{aligned}
P\left(\left|(\Delta W_\star^{(m)} u)_m^{ij}\right| \geq t\right) &\leq P\left(\left|\sum_{k=1}^n e_{mk} \sum_{s=1}^{d_k} \bar{W}_{mk}^{is} u_k^{sj}\right| \geq t\right) + O(n^{-5}) \\
&= P\left(\left|\sum_{k=1}^n e_{mk} \sum_{s=1}^{d_k} \bar{W}_{mk}^{is} u_k^{sj}\right| \geq t, A\right) + P\left(\left|\sum_{k=1}^n e_{mk} \sum_{s=1}^{d_k} \bar{W}_{mk}^{is} u_k^{sj}\right| \geq t, A^c\right) + O(n^{-5}) \\
&\leq P\left(\left|\sum_{k=1}^n e_{mk} \sum_{s=1}^{d_k} \bar{W}_{mk}^{is} \bar{u}_k^{sj}\right| \geq t\right) + O(n^{-3}) + O(n^{-5}) \\
&= P\left(\left|(\Delta \bar{W}_\star^{(m)} \bar{u})_m^{ij}\right| \geq t\right) + O(n^{-3}).
\end{aligned}
\tag{179}
$$

Now, we bound $P\left(\left|(\Delta \bar{W}_\star^{(m)} \bar{u})_m^{ij}\right| \geq t\right)$. Since $\Delta W_\star^{(m)}$ is independent from $u$, $\Delta \bar{W}_\star^{(m)}$ is also independent from $\bar{u}$. Therefore, as argued in the proof of lemma 3.4, we can condition on $\bar{u}$ and view $\bar{u}$ as deterministic. In this case, $\{e_{mk} \sum_{s=1}^{d_k} \bar{W}_{mk}^{is} \bar{u}_k^{sj}\}_{k \in [n]}$ are $n$ mean zero, independent random variables.

$$\left|e_{mk} \sum_{s=1}^{d_k} \bar{W}_{mk}^{is} \bar{u}_k^{sj}\right| \leq \sum_{s=1}^{d_k} |\bar{W}_{mk}^{is}| \cdot |\bar{u}_k^{sj}| \leq c' \bar{d} n^{\frac{1}{4}} \tag{180}$$

Besides, we have

$$\sum_{k=1}^n \mathbb{E}\left((e_{mk} \sum_{s=1}^{d_k} \bar{W}_{mk}^{is} \bar{u}_k^{sj})^2\right) = p \sum_{k=1}^n \sum_{s=1}^{d_k} (\bar{u}_k^{sj})^2 \mathbb{E}(\bar{W}_{mk}^{is})^2 \leq arnp, \tag{181}$$

Using lemma H.2, we derive

$$P(|\sum_{k=1}^n e_{mk} \sum_{s=1}^{d_k} \bar{W}_{mk}^{is} \bar{u}_k^{sj}| \geq t) \leq 2 exp\left(-\frac{t^2/2}{a^2 rnp + c\bar{d} n^{\frac{1}{4}} \frac{t}{3}}\right). \tag{182}$$

Letting $t = c\sqrt{np \log n}$, together with the condition $p > c\frac{\log n}{\sqrt{n}}$, we have with probability at least $1 - O(n^{-3})$, $\|(\Delta \bar{W}_\star^{(m)} \bar{u})_m\|_F \lesssim \sqrt{np \log n}$. Therefore, with probability at least $1 - O(n^{-3})$, $\|(\Delta W_\star^{(m)} u)_m\|_F \lesssim \sqrt{np \log n}$.

**Step 2:** For the second term, we use bound $\|u_m\|_F \leq \frac{8\sqrt{r}}{7}$ in lemma H.7 to modify (146).

$$\sum_{k \neq m} \|(\Delta W_\star^{(m)} u)_k\|_F^2 = \sum_{k \neq m} \|(W_\star)_{km} u_m\|_F^2 \leq \|u_m\|_F^2 \sum_{k \neq m} e_{mk} \|W_{km}\|_F^2 \tag{183}$$

Since (150), $P(\sum_{k \neq m} e_{mk}\|W_{km}\|_F^2 \geq c_5 np) = O(n^{-5})$.

$$
\begin{aligned}
P\left(\|u_m\|_2^2 \sum_{k \neq m} e_{mk}\|W_{km}\|_F^2 \geq \frac{64r}{49}c_5 np\right) &= P\left(\|u_m\|_F^2 \sum_{k \neq m} e_{mk}\|W_{km}\|_F^2 \geq \frac{64r}{49}c_5 np, \|u_m\|_F \leq \frac{8}{7}\sqrt{r}\right) \\
&\quad + P\left(\|u_m\|_F^2 \sum_{k \neq m} e_{mk}\|W_{km}\|_F^2 \geq \frac{64r}{49}c_5 np, \|u_m\|_F > \frac{8}{7}\sqrt{r}\right) \\
&\leq P\left(\sum_{k \neq m} e_{mk}\|W_{km}\|_F^2 \geq c_5 np\right) + P\left(\|u_m\|_F > \frac{8}{7}\sqrt{r}\right) \\
&\lesssim n^{-3}.
\end{aligned}
\tag{184}
$$

Therefore, with probability at least $1 - O(n^{-3})$, $\sum_{k \neq m}\|(\Delta W_\star^{(m)} u)_k\|_F^2 \lesssim np$. Combing step 1 and step 2 and taking a union bound over $m \in [n]$, we complete the proof. $\qquad\square$

**Lemma H.9.** *Let $\sigma \lesssim \sqrt{np/\log n}$, $p > c\frac{\log n}{\sqrt{n}}$, Then with probability at least $1 - O(n^{-2})$, $\forall m \in [n]$ we have*

$$
\|\Delta(\Delta_\star)^{(m)}\widetilde{G}^{(m)}\|_F \leq C_5\sqrt{np\log n}
\tag{185}
$$

*Proof.* We only need to modify the proof of lemma H.5 with the bound in lemma H.7. Let $u = \widetilde{G}^{(m)}$

$$
\|\Delta(\Delta_\star)^{(m)}u\|_F^2 = \|(\Delta(\Delta_\star)^{(m)}u)_m\|_F^2 + \sum_{k \neq m}\|(\Delta(\Delta_\star)^{(m)}u)_k\|_F^2
\tag{186}
$$

**Step 1:** For the first term, we define $\bar{u}$ and event $A$ as in the proof of lemma H.8.

$$
\begin{aligned}
P\left(\left|\Delta(\Delta_\star)^{(m)}u)_m^{ij}\right| \geq t\right) &= P\left(\left|\Delta(\Delta_\star)^{(m)}u)_m^{ij}\right| \geq t, A\right) + P\left(\left|\Delta(\Delta_\star)^{(m)}u)_m^{ij}\right| \geq t, A^c\right) \\
&\leq P\left(\left|\Delta(\Delta_\star)^{(m)}\bar{u})_m^{ij}\right| \geq t\right) + O(n^{-3}) \\
&= P\left(\left|\sum_{k=1}^n \left((e_{mk} - p)\sum_{s=1}^{d_k}(G_m^\star G_k^{\star\top})^{is}\bar{u}_k^{sj}\right)\right| \geq t\right) + O(n^{-3}).
\end{aligned}
\tag{187}
$$

Now, we bound $P\left(\left|\Delta(\Delta_\star)^{(m)}\bar{u})_m^{ij}\right| \geq t\right)$. Since $u$ is independent from $\Delta(\Delta_\star)^{(m)}$, $\bar{u}$ is also independent from $\Delta(\Delta_\star)^{(m)}$. Therefore, we can condition on $\bar{u}$ and view $\bar{u}$ as deterministic matrix. In this case, $\{(e_{mk} - p)\sum_{s=1}^{d_k}(G_m^\star G_k^{\star\top})^{is}\bar{u}_k^{sj}\}_{k \in [n]}$ are $n$ mean zero, independent random variables.

$$
\left|(e_{mk} - p)\sum_{s=1}^{d_k}(G_m^\star G_k^{\star\top})^{is}\bar{u}_k^{sj}\right| \leq \sum_{s=1}^{d_k}\|(G_m^\star G_k^{\star\top})\|_F \cdot \|\bar{u}_k\|_F \leq C_4\bar{d}^{\frac{3}{2}}r\frac{n^{\frac{1}{4}}}{\sqrt{\log n}}
\tag{188}
$$

Besides, through the same calculation as in (155), we have $\sum_{k=1}^n \mathbb{E}\left((e_{mk} - p)\sum_{s=1}^{d_k}(G_m^\star G_k^{\star\top})^{is}\bar{u}_k^{sj}\right)^2 \leq r^2 np$. Using lemma H.2, we have

$$
P\left(\left|\sum_{k=1}^n \left((e_{mk} - p)\sum_{s=1}^{d_k}(G_m^\star G_k^{\star\top})^{is}\bar{u}_k^{sj}\right)\right| \geq t\right) \leq 2exp\left(-\frac{t^2/2}{r^2 np + C_4\bar{d}^{\frac{3}{2}}r\frac{n^{\frac{1}{4}}}{\sqrt{\log n}}\frac{t}{3}}\right).
\tag{189}
$$

Taking $t = c\sqrt{np\log n}$, we have that with probability $1 - O(n^{-3})$, $\|(\Delta(\Delta_\star)^{(m)}\bar{u})_m\|_F \lesssim \sqrt{np\log n}$. Since (187), with probability at least $1 - O(n^{-3})$, $\|(\Delta(\Delta_\star)^{(m)}u)_m\|_F \lesssim \sqrt{np\log n}$.

**Step 2:** For the second term,

$$\sum_{k \neq m} \|(\Delta(\Delta_\star)^{(m)} u)_k\|_F^2 = \sum_{k \neq m} \|(\Delta_\star)_{km} u_m\|_F^2 \leq \|u_m\|_F^2 \sum_{k \neq m} \|(\Delta_\star)_{km}\|_F^2$$

$$= \|u_m\|_F^2 \sum_{k \neq m} (e_{km} - p)^2 \|G_k^\star G_m^{\star\top}\|_F^2 \leq r \|u_m\|_F^2 \sum_{k=1}^n (e_{mk} - p)^2. \tag{190}$$

Since the bound $\sum_{k=1}^n (e_{mk} - p)^2 \lesssim np$ in the proof of lemma H.5 still holds, using a similar decomposition as in (184), we have that with probability at least $1 - O(n^{-3})$, $\sum_{k \neq m} \|(\Delta(\Delta_\star)^{(m)} u)_k\|_F^2 \lesssim np$.

Combing step 1 and step 2 and taking a union bound over $m \in [n]$, we get the desired result. $\square$

## I. Proof of Theorem 3.11

Using lemma H.8 and lemma H.9, we have $\forall m \in [n]$,

$$\|\Delta \widetilde{W}^{(m)} \widetilde{G}^{(m)}\|_F \leq \frac{1}{p} \|\Delta(\Delta_\star)^{(m)} \widetilde{G}^{(m)}\|_F + \frac{\sigma}{p} \|\Delta W_\star^{(m)} \widetilde{G}^{(m)}\|_F \lesssim n. \tag{191}$$

Using lemma H.4 and lemma H.5, we have

$$\forall m \in [n], \forall t \leq 3n^2, \|\Delta \widetilde{W}^{(m)} G^{t,m}\|_F \lesssim n, \tag{192}$$

where $G^{t,m}$ is the sequence generated by GPM algorithm with $C$ replaced by $C_p$. Using lemma H.1 we have

$$\|\widetilde{W}\|_2 \lesssim n, \quad \|\widetilde{W}^{(m)}\|_2 \lesssim n, \quad \|\Delta \widetilde{W}^{(m)}\|_2 \lesssim n, \quad \|\widetilde{W}_m\|_2 \lesssim n. \tag{193}$$

The above three properties and the fact $C_p = G^\star G^{\star\top} + \widetilde{W}, C = G^\star G^{\star\top} + \sigma W$ imply that $\widetilde{W}$ is at the exactly same status as $\sigma W$. Therefore, the result in the full observation model can be generalized to the partial observation case easily.

