# OpenReview forum: "Nearly Optimal Bounds for Orthogonal Trace-Sum Maximization"
_ICML.cc/2026/Conference — Submitted to ICML 2026_

### Official Review · Reviewer_kqMW · 2026-02-16

**Soundness:** 2
**Presentation:** 3
**Significance:** 2
**Originality:** 1
**Overall Recommendation:** 3
**Confidence:** 3

**Summary:**

This paper studies orthogonal trace-sum maximization (OTSM) on Stiefel manifolds, motivated by applications such as generalized CCA. Under a planted signal plus Gaussian noise model for the block matrix of pairwise measurements, the authors analyze two algorithmic approaches that are standard in the literature: (i) a semidefinite relaxation (SDR) of OTSM and (ii) the generalized power method (GPM) with spectral initialization. The main claim is that, with high probability, both methods succeed under an improved noise regime, closing the gap between prior theory and empirical behavior.

**Compliance With Llm Reviewing Policy:**

Affirmed.

**Final Justification:**

Thank you for addressing my more minor comments and for the additional clarifications. As my major comments are about the significance of the work and not easily addressable in a revision, I am keeping my score unchanged.

**Key Questions For Authors:**

1. Do the authors believe they have proved Theorem 3.11? The appendix proves three related properties and then reads “Therefore, the result in the full observation model can be generalized to the partial observation case easily.” However, there are several key steps from the full observation case which are missing in the partial observation case, so I am not convinced that the result has been proven in the partial observation case.
2. Are the authors able to produce families of instances which attain their bounds for their main results? In particular, for Theorem 1, is it possible to produce a family of instances where \sigma=\Omega(sqrt(np/log n)) and the SDP relaxation is tight, but for the same family it is not tight for \Omega((np/log n)^{0.5+\epsilon})? I would find the analysis more convincing if the authors could show that the new bounds are tight, and not an artifact of their proof techniques
3. Can the authors please explain the use of the word “unique” in the introduction? The objective function is invariant under right multiplying all blocks by a common rotation matrix. Same question for the statement “then O is the unique global maximizer” in Lemma 3.9
4. In section 4, the authors describe their experimental setup by writing “according to the Gaussian noise model (5).” Could the authors elaborate on what the noise model is, e.g., what is the covariance matrix?
5. In Equation (30), where does the \sqrt{r} term come from? I get \sqrt{n} not \sqrt{nr} when bounding the term in (30) [recall that ||AB|\_F <= ||A||_F ||B||_2, not just <= ||A||_F ||B||_F]. If I am right, does that impact any of the results?

**Limitations:**

I could not find a limitations discussion in the paper

**Strengths And Weaknesses:**

Strengths: the paper improves the noise regime compared to Won et al. (2021) and the main claims in the paper appear to be correct.

Weaknesses: this paper reads as a bag of ideas about different algorithms, rather than a single unified story, and I believe that none of the improvements in the analysis of the algorithms are individually significant enough for ICML. The work of Bandeira et al. (2016) already provides a quite tight analysis of the semidefinite relaxation studied in this paper without any assumptions on the objective matrix; the main contribution in this paper is to provide some sufficient conditions for when the SDP relaxation is exact, under some quite strong conditions on the generative model that cannot be checked in practice. Similarly, the improved constant in the “nearly optimal bounds” contribution (C.2) seems to be quite a marginal improvement over existing literature.

In addition, the paper seems to depend a lot on Won et al. (2021). For instance, the first page of the paper reads very similarly to the first page of Won et al. (2021), the examples of when the problem arises in practice (Section 2) refers to Won et al. (2021) for more details, the optimality conditions are adapted from Won et al. (2021), and the overall problem setup/SDP relaxation idea/certification style look like a straightforward continuation of Won et al. (2021). This makes me think that the paper is too incremental for a top conference like ICML.

Finally, I believe that the authors should explicitly mention the guarantees provided by the works of Naor et al. (2014) “Efficient Rounding for the Noncommutative Grothendieck Inequality” which gives a 1/2sqrt(2) approximation algorithm for a more general class of problems which encompasses the problem studied here, as well as Bandeira et al. (2016) “Approximating the little Grothendieck problem over the orthogonal and unitary groups” as both works provide guarantees independent of the data generation process 1/2sqrt(2) and something at least as strong, respectively).

---

> ### Author Rebuttal · Authors · 2026-03-30
>
> ## Answer to Key Questions
> ### 1.Correctness of proof for Theorem 3.11
> Regarding this proof, we argued for its correctness at the beginning of Section 3.2. Firstly, we refomulate the partial observation model as $$C_p=G^\star G^{\star\top}+\tilde{W}$$ (19), where $\tilde{W}=\frac{1}{p}\Delta_\star+\frac{\sigma}{p}W_\star$. Recall that the full observation model is given by $C=G^\star G^{\star \top}+\sigma W$. The two models are identical **except for the difference in the noise terms**. In the proof of the full observation model, all  requirements regarding the noise term $\sigma W$ are contained in Lemma A.1 and Lemma 3.4. The rest of the analysis are independent of the explicit form of noise term. In the proof of Theorem 3.11, we establish Lemma H.1 and Lemma 3.12, which are identical to Lemma A.1 and Lemma 3.4. With these two lemmas, Theorem 3.11 can be proved simply by replacing all instances of $\sigma W$ appearing in the full observation model proof with $\tilde{W}$.
>
> ### 2. Empirical evidence for tightness of our bound for $\sigma$
>  We conduct the following experiment to check the tightness of SDP relaxation. Due to  time limit, the experiment is conducted on small size problems. The following table summarizes the ratio (out of 10 instances) of tight relaxation.
> | $\sigma / \sqrt{n}$ | n=20 | n=40 | n=60 | n=80 | n=100 |
> | :--- | :--- | :--- | :--- | :--- | :--- |
> | **0.155** | 1.0 | 1.0 | 1.0 | 1.0 | 1.0 |
> | **0.160** | 0.7 | 0.8 | 0.6 | 0.4 | 0.2 |
> | **0.165** | 0.3 | 0.0 | 0.2 | 0.1 | 0.0 |
> | **0.170** | 0.2 | 0.1 | 0.1 | 0.0 | 0.0 |
>
> It can be seen that a jump happens between 0.155 and 0.165, which matches the ratio 0.158 shown in Figure 1. We will conduct larger scale experiment in our revision.
>
> ### 3. Explaination of uniquness
>  As illustrated in Definition 2.1, all concepts of convergence and uniqueness are defined in the quotient space, i.e., up to a global rotation.
>
> ### 4. Clarification of  Gaussian noise model
> (5)  is identical to (2) with $\Omega=[n] \times [n]$. $W$ is a  symmetrical $nd \times nd$ matrix, where the entries on and above the diagonal are independent standard Gaussian random variables.
>
> ### 5. Typos in equation (30)
> Yes, $\sqrt{r}$ in equation (30) can be eliminated. Thank you for pointing out this.
>
> ## Answer to Weakness
> ### 1. Relationship between our work and Bandeira et al. (2016)[1]
> Bandeira et al. (2016)[1] prove a $\frac{1}{2\sqrt{2}}$ lower bound for SDP relaxation without  assumption on the generative model. However, this is still far from  tightness of the SDP and is insufficient for practical use. By contrast, we not only prove  exact tightness of SDP but also prove the linear convergence of GPM, which is far more efficient than SDP for large-scale problems.
>
> Besides, our generative model (2) arises naturally in many practical application scenarios, such as CCA and cryo-EM. It is also standard in the literature, used by Bandeira et al. (2017)[2] and Won et al. (2022)[4].
>
> ### 2. Relationship between our work and Won et al. (2021)[3]
>
> Won et al. (2021)[3]  only provide a criterion for checking optimality, but do not prove that SDP satisfies such criterion. Hence, SDP tightness is not guaranteed. Won et al. (2022)[4]  consider data generation model $S=\Theta \Theta^\top + \sigma W$ and prove tightness of  SDP under  $O(n^{\frac{1}{4}})$ noise level, but this result is suboptimal regarding empirical observation.
>
> Our main contribution compared to Won et al. (2022)[4] is the **improvement of the noise level requirement from $O(n^{\frac{1}{4}})$ to nearly optimal level $O(\sqrt{n/ \log n})$**. This improvement stems from our leave-one-out proof framework. As explained in Section 3.1.1 of our article, this technique resolves the intrinsic dependence of  GPM iteration $G^t$ on the noise matrix $W$ and provides $O(\sqrt{n \log n})$ bound for $\\|WG^t\\|_{\infty}$, which is quite non trivial.
>
> ### 3. Unification of our article
> In this paper, we study the noise tolerance of a specific class of OSTM problems— the synchronization problem on the Stiefel manifold . To this end, we consider two SOTA algorithms: GPM and SDP.
>
> We primarily focus on proving  linear convergence of  GPM under a noise intensity of $O(\sqrt{n/\log n})$ via Leave-One-Out technique. As a corollary of  GPM’s convergence, we obtain tightness of  SDP by verifying Lemma 3.9 for $O=G^\infty$.  Therefore, the two algorithms  are not isolated; their analyses are closely linked.
>
> [1]: Approximating the little Grothendieck problem over the orthogonal and unitary groups，Afonso S. Bandeira1 et al.
>
> [2]: On the low-rank approach for semidefinite programs arising in synchronization and community detection,Afonso S. Bandeira et al.
>
> [3]:Orthogonal trace-sum maximization: applications, local algorithms,and global optimality, Joong-ho Won  et al.
>
> [4]:Orthogonal trace-sum maximization: tightness of the semidefinite relaxation and guarantee of locally optimal solutions,Joong-ho Won et al.

---

> > ### Author Rebuttal · Reviewer_kqMW · 2026-04-02
> >
> > Thank you for addressing my more minor comments and for the additional clarifications. As my major comments are about the significance of the work and not easily addressable in a revision, I am keeping my score unchanged.

---

### Official Review · Reviewer_sb1C · 2026-03-05

**Soundness:** 3
**Presentation:** 2
**Significance:** 3
**Originality:** 3
**Overall Recommendation:** 4
**Confidence:** 3

**Summary:**

This paper presents new theoretical results on recovery guarantees for the Orthogonal Trace-Sum Maximization (OTSM) problem. Semidefinite relaxation (SDR) and the generalized power method (GPM) have been the two primary algorithmic approaches to this problem. The paper assumes a Gaussian noise model, in which ground-truth signals $G_i^\star$ are observed through additive Gaussian noise matrices $W_{ij}$ with noise level $\sigma$.

The paper makes the following contributions:

1. It establishes linear convergence of GPM to a global optimum in the fully observed case ($p=1$), provided that $\sigma = O(\sqrt{n / \log n})$. The authors also empirically demonstrate the convergence behavior of GPM.
2. It provides improved theoretical guarantees for the tightness of SDR in the noise regime $\sigma = O(\sqrt{n / \log n})$, improving upon the previous bound of $\sigma = O(n^{1/4})$.

**Compliance With Llm Reviewing Policy:**

Affirmed.

**Key Questions For Authors:**

1. The model assumes the Gaussian noise, and Erd˝os–R´enyi observation model, which seems to be ideal. Could you explain any realistic scenarios or applications where this setting can be applied?

**Limitations:**

yes

**Strengths And Weaknesses:**

- Strengths

The main strength of the paper is its substantial improvement over prior results. For GPM, the paper establishes linear convergence to a global optimum under the noise regime $\sigma = O(\sqrt{n / \log n})$. For SDR, it proves recovery guarantees under $\sigma = O(\sqrt{n / \log n})$ both in the fully observed setting ($p = 1$) and under random observations.

The experiments also provide empirical support for the paper’s theoretical claims.

- Weaknesses

The main weakness is the lack of lower-bound or impossibility results in the high-noise regime. There appears to be empirical evidence, as well as similarities to special cases such as angular synchronization and phase synchronization, suggesting that $\sigma = \tilde{\Omega}(\sqrt{n})$ may be an information-theoretically impossible regime for recovery, but the paper does not discuss such results.

In addition, the proofs seem overly technical in their current presentation. Providing a high-level overview or proof intuition would likely improve the paper’s readability.

---

> ### Author Rebuttal · Authors · 2026-03-30
>
> ## Answer to Key Questions
> First, we want to point out that the Gaussian noise assumption in our article can be directly relaxed to sub-Gaussian for the full observation model and symmetric sub-Gaussian (density function $p$ satisfies $p(x)=p(-x)$ ) for the random observation model. In our proof, we only use concentration inequalities such as Lemma H.3  that generally hold for all sub Gaussian random variables and we  do not use any special property of the Gaussian distribution.  We will change our assumption for the noise term to sub Gaussian in our final version of paper. Besides, although  Erdos–Renyi observation model seems ideal, it is  standard in many literature analysing similar problems.
>
> As for application scenarios,  model (2) is natural for generalized CCA in Won et al. (2022)[1]. Given several  high-dimensional random vectors, CCA tries to find their common latent random vector in a low-dimensional space that best explains their common characteristics.
>
> Consider $n$ high-dimentional random vectors $a_i \in R^{d_i},i \in [n]$ driven by a common latent vector $z\in\mathbb{R}^r$:
> $$a_i = G_i^* z + \epsilon_i$$
> where $G_i^\star \in O_{d_i,r}$ (non-random) form an orthogonal basis for an $r$-dimensional subspace of $R^{d_i}$. $\epsilon_i$ has mean zero and is uncorrelated with $z$ and $\epsilon_j$ for $j \ne i$. We further assume that the covariance  of $\epsilon_i$ is $\tau I_{d_i}$ and that $z$ has zero mean and covariance  $I_r$.  The goal of CCA is to infer $G^{\star}=\\{G_i^\star\\}\_{i \in [n]}$  by observing  $\\{a_i\\}\_{i \in [n]}$. To this end, for each $a_i$, we draw an i.i.d sample  denoted by $a_i^k=G_i^\star z^k + \epsilon_i^k, k \in [m]$ and define $A_i=[a_i^1,\cdots, a_i^m]$.    Since
>
> $$
> E[\frac{1}{m}A_iA_j^\top]= \begin{cases}
> G_i^\star G_j^{\star\top} & i \ne j, \\\\
> G_i^\star G_i^{\star\top}+\tau I_{d_i} & i=j,
> \end{cases}
> $$
>
> we define $W_{ij}=\frac{1}{m}A_iA_j^\top-E[\frac{1} {m}A_iA_j^\top]$. Then the MAXBET problem:
>
> $$
> \begin{aligned}
> \max \quad & \sum_{i, j} \operatorname{tr}\left(O_{i}^{\top}A_iA_j^\top O_j\right) \\\\
> \text{s.t.} \quad & O_{i} \in O_{d_i, r} , \quad i \in [n],
> \end{aligned}
> $$
>
> is equivalent to an OSTM problem with $C_{ij}=G_i^\star G_j^{\star\top}+W_{ij}$, which will recover $G^\star$ up to some global rotation. We will add this discussion to our revision.
>
> Besides, the $C=G^\star G^{\star \top}+\sigma W$ structure also arises in other applications such as cryo-EM[2] and multiview structure from motion[3].
>
> ## Answer to weakness
> ### 1. Impossibility analysis
> As for impossibility analysis,  information-theoretic lower bounds for $\sigma$ only exist for subgroups of $O(d)$ , such as $Z_2$[4], $S_N$[5], $SO(2)$[6], $SO(3)$[5], which can not be directly applied to Stiefel manifold.  Establishing such lower bound is heavily nontrivial, and we leave this as our future work.
>
> Although we lack a theoretical impossibility result, Figure 1  provides empirical evidence. The red boundary line  indicates the failure of GPM  when the noise level exceeds $O(\sqrt{n})$.
>
> ### 2. High level overview of our proof.
> We prove for two main results  under the $O(\sqrt{n/ \log n})$ noise level, which are the linear convergence of GPM and tightness of SDP relaxion.
>
> For convergence of GPM, we use the standard contractive map arguement.
> - We first show that spectral initialization gives a point $\tilde{G}$  very close to  $G^\star$ both in $d_F$ and $d_\infty$ distance. (Theorem 3.3)
> - Secondly, we show that the GPM iteration map $\mathcal{P}_C$ has local contractive property in a small neiborhood of $G^\star$. (Lemma 3.5)
> - Then, we show that the whole sequence of GPM iteratiom stays in the neiborhood described in Lemma 3.5 (Theorem 3.7). To this end, we need to derive $O(n\log n)$ bound for $\\|W G^t\\|_\infty$. Therefore, we use the leave-one-out technique and introduce several auxiliary sequences $G^{t,m}$ to solve the intrinsic  dependence of $G^t$ on noise $W$ .
> - Equipped with the above properties, GPM iteration can be seen as a contractive map along the iteration steps $G^t$, which implies linear convergence of GPM
> - The tightness of SDP is given as a direct corrolary of the convergence of GPM by verifying Lemma 3.9 for $O=G^\infty$.
>
> We will include an overall proof roadmap in our final version of paper.
>
> [1]: Orthogonaltrace-sum maximization:tightness of the semidefinite relaxation and guarantee of locally optimal solutions,Joong-ho Won et al.
>
> [2]: Viewing direction estimation in Cryo-EM using synchronization, Yoel Shkolnisky et al.
>
> [3]: Global motion estimation from point matches, M. Arie-Nachimson et al.
>
> [4]: Decoding binary node labels from censored edge measurements: Phase transition and efﬁcient recovery, Abbe et al.
>
> [5]: Information recovery from pairwise measurements, Chen et al.
>
> [6]: Angular synchronization by eigenvectors and semideﬁnite programming, Singer

---

> > ### Author Rebuttal · Reviewer_sb1C · 2026-04-03
> >
> > The authors have addressed my concerns in the rebuttal adequately. I remain positive about the paper.

---

### Official Review · Reviewer_Rai8 · 2026-03-11

**Soundness:** 3
**Presentation:** 3
**Significance:** 2
**Originality:** 3
**Overall Recommendation:** 5
**Confidence:** 2

**Summary:**

This paper studies the Orthogonal trace-sum maximization (OTSM) problems and provides theoretical guarantees for both the semidefinite programming (SDP) relaxation method and the generalized power method (GPM). Authors show that, starting with spectral initialization, GPM converges linearly to global optimal solution with high probability if noise is $\mathcal O(\sqrt{n / \log n})$, and the relaxation of SDP is tight under the same noise level. They further extend the result to E-R model for randomly observable settings. When $p\gtrsim \log n /\sqrt{n}$, both linear convergence of GPM and tightness of SDP relaxation holds for $\mathcal O(\sqrt{n p/ \log n})$ noise. The noise level is nearly optimal, which match the empirical result of $\mathcal O(\sqrt{n})$.

**Compliance With Llm Reviewing Policy:**

Affirmed.

**Final Justification:**

My concerns are resolved.
But I'm not sure about the concern raised by Reviewer kqMW.
It seems that the past literature were not thorougly discussed and the theoretical contribution is incremental.
I still maintain "accept" but I'm not quite familiar with all past bounds.

**Key Questions For Authors:**

- In section 4.2, it seems that $\frac{\sqrt{p}}{\sigma} d_F\left(\widehat{G}^{\infty}, G^{\star}\right)/ \frac{\sqrt{p}}{\sigma} d_F\left(G^0, G^{\star}\right)$ remain a constant very close to 1, does this indicates that the GPM estimator has a neglible improvement on the spectral initialization? It seems contradict with the linear convergenve result.
- The spectral initialization will construct a good solution close to the optimal solution, will it be possible for the good preperty of region of contraction be applied to other non-convex methods rather than power iteration, to avoid the SVD cost for each iteration? What is the practical problem size for real-world OTSM problems?

**Limitations:**

- The constant in theoretical proof is extremely conservetive.
- This work relies on the additive Gaussian noise model, and real-world applications may require other sub-gaussian/block-wise/heavy tailed ones.
- One typo: Eqn 50 & 53, additional )

**Strengths And Weaknesses:**

Strength:
- This paper uses the Leave-One-Out technique and extend the $\mathcal O(\sqrt{n / \log n})$ noise bound to OTSM problem rather than orthogonal group synchronization problem where $d_i=r$. It improves the sub-linear convegrence rate and $\mathcal O(n^{1/4})$ in (Won 2021, 2022) in the fully observed setting.
- It connects the duality result of SDP with the convergence result of GPM, claiming the uniqueness and global optimality for the GPM method, and all the above bound can be extended to random graph model.
- The proof idea is well-written in the main-body, easily understood without checking much of the appendix.

Weakness:
- The main proof framwork follows the LOO technique and those in orthogonal group synchronization paper (a special case if OTSM), wondering if it's possible to discuss the influence of d on the bound, like $\sqrt{n} / (\sqrt{d} + \sqrt{\log n})$ from the numerical result.

---

> ### Author Rebuttal · Authors · 2026-03-30
>
> ## Answer to key questions
> ### 1. Explanation for the small improvement of the GPM estimation compared to the spectral initialization.
>
> - In the ideal case where the observation probability $p$ is large enough and the noise level $\sigma$ is relatively small, these two distances are not expected to be very different, as both $d_F(G^0,G^\star)$ and $d_F(G^\infty,G^\star)$ have an upper bound of order $O(\frac{1+\sigma}{\sqrt{p}})$. This phenomenon is commonly seen in literature discussing  synchronization problems.
>
> - However, after choosing a smaller observation probability $p$, which is $p=0.05\frac{\log n}{\sqrt{n}}$, we find a significant improvement of GPM over spectral initialization. Experiment results are shown in the following table：
> | $n$ | 100 | 200 | 300 | 400 | 500 |
> | :--- | :---: | :---: | :---: | :---: | :---: |
> | $\frac{\sqrt{p}}{\sigma}d_F(G^0,G^\star)$ | 7.12 |5.37 | 4.80 | 4.46 | 4.23 |
> | $\frac{\sqrt{p}}{\sigma}d_F(G^\infty,G^\star)$ | 6.05| 4.48 | 3.99 | 3.77 | 3.64 |
> | Improvement Rate | 15%| 17% | 17% | 15% | 14% |
>
> - This phenomenon does not contradict with the linear convergence of GPM , which is shown in Figure 4. Because of the noise term $W$, the optimal solution $O^\star$ of OSTM is not equal to the ground truth $G^\star$. Therefore, although $d_F(\widehat{G}^\infty,O^\star)$ is much smaller than $d_F(G^0,O^\star)$, it is possible that $d_F(G^0,G^\star)$ and $d_F(G^\infty,G^\star)$ are very close.
>
> ### 2. Other non-convex methods to avoid SVD.
> There are some non-convex methods discussed in the literature that can avoid SVD. Bin et al. (2022)[1] discuss an optimization method on the Stiefel manifold.  This method starts from a point that is not on the manifold and gradually 'lands' on the manifold while approaching the optimum. This method does not need to keep the iteration points on the manifold and therefore avoids SVD. Bin et al. (2020)[2] proposed a PLAM method in Algorithm 1 that handles constraints using augmented Lagrangian function and avoids projection. We may try to study these algorithms in our future work.
>
> Besides, we want to point out that in real application scenarios of OSTM, SVD decomposition is usually not a serious problem. For example, in cryo-EM, we solve an $SO(3)$ synchronization problem, which means that we only need to perform SVD for $3 \times 3$ matrices, and the SVD operations in one iteration can be done in parallel. By contrast, in this problem, $n$ is very large, usually reaching $10^{5}$. Since the noise level for cryo-EM is very high (with SNR lower than 0.1), it is more crucial for an algorithm to have high tolerance for noise.
>
>
> ## Answer to weakness
> After carefully checking our proof, we derive that the influence of $d$ on the bound is $\sigma \lesssim \sqrt{n /(d \log n)}$ (  $r$ viewed as an absolute constant).
>
> We will design numerical experiments similar to that in Figure 1  to draw a heat map between $\sigma$ and $d$. Since this experiment will take several days to complete, we plan to include this in our final paper.
>
> ## Answer to limitations
> ### 1. Conservative constants in the proof
> In our bound $\sigma \leq C\sqrt{n/ \log n}$, our choice of constant $C$ may be overly conservative. However, this constant is not intrinsic and can be adjusted in the proof. For example, if we relax the probability estimation from $1-O(n^{-2})$ to $1-O(n^{-1})$, then the constant $C$ can be larger.
>
> ### 2. Gaussian noise assumption
> We want to point out that the Gaussian noise assumption in our article can be directly relaxed to sub-Gaussian for the full observation model and symmetric sub-Gaussian (the density function $p$ satisfies $p(-x) = p(x)$) for the random observation model. In our proof, we only use concentration inequalities such as Lemma H.3  that generally hold for all sub Gaussian random variables and we  do not use any special property of the Gaussian distribution. We will change our assumption for the noise term to sub Gaussian in our final version of paper.
>
> [1]: Optimization ﬂows landing on the Stiefel manifold Bin Gao, Simon Vary, Pierre Ablin, P.-A. Absil
>
> [2]:An orthogonalization-free parallelizable framework for all-electron calculations in density functional theory bin gao, guanghui hu, yang kuang and xin liu

---

> > ### Author Rebuttal · Reviewer_Rai8 · 2026-04-03
> >
> > Thanks authors for their comprehensive responses. My concerns are resolved. But I'm not sure about the concern raised by Reviewer kqMW. I will maintain "accept" as the $\mathcal O(\sqrt{n/\log n})$ bound seems still new to OTSM problem.

---

### Official Review · Reviewer_nsWS · 2026-03-12

**Soundness:** 3
**Presentation:** 2
**Significance:** 3
**Originality:** 3
**Overall Recommendation:** 5
**Confidence:** 2

**Summary:**

The article provides theoretical analysis for the multi-block version of the orthogonal trace-sum maximization problem:

$\sum_{i,j} Tr(O_i^T C_{ij} O_j)$

where optimization variables $O_i$'s are in Stiefel manifold.

The article considers two algorithmic approaches: 1. Semidefinite Relaxation 2. Generalized Power Method

Main question addressed is the convergence behaviour of GPM algorithm when SDR approximation is tight.

Under a stochastic generative model on optimization parameters $C_{ij}$ (with Gaussian noise), the article shows the linear convergence of GPM algorithm to global optima and the tightness of the SDP relaxation under the low noise regime.

**Compliance With Llm Reviewing Policy:**

Affirmed.

**Final Justification:**

Based on the rebuttal explanations of the authors to all reviewers' comments, I am more convinced that this article offers a frontier result in optimization analysis, especially the  $\cal{O}(\sqrt{n/log(n)})$ bound.

**Key Questions For Authors:**

could you provide more motivation for the generative model for the observation matrices in (2).
in connection, it would be helpful to have a discuss why is the bound on sigma is critical.

**Limitations:**

Yes

**Strengths And Weaknesses:**

STRENGTHS
1. The article provides strong theoretical results about the global convergence for a highly non-convex problem. t seems the article change the noise condition from O($n^{1/4}$) in  Won et.al 2022 to O($\sqrt{n/\log(n)}$, proving linear convergence for GPM. However, due to my unfamiliarity with the existing literature in this area, I am not able to judge the relative contribution level of the presented work.i)

2. The simulations at the end of the article are informative and they nicely complement the theoretical findings.


WEAKNESS
I believe the presentation could be improved to make the article more accessible and clear.

The introduction could provide more motivation for the OTSM problem.

There exists point in the article that require clarification, e.g.,

- the inner product in SDP  in (1)
-  C matrix in SDP  in (1)
-  How U and r relates to the original OTSM setup
-  The motivation behind the ground truth in (2)

---

> ### Author Rebuttal · Authors · 2026-03-30
>
> ## Clarification of notations and explanation of (1)
> 1. Clarification of notations
> - The inner product in (1) is defined by $\langle A,B \rangle=tr(AB^\top)$.
> - Matrix $C$ in (1) is a block matrix with $(i,j)$-th block equal to $C_{ij}$ for  $(i,j) \in \Omega$, and  0 for $(i,j) \notin \Omega$.
>
> 2. Relationship between OSTM and the relaxed problem (1)
>
> We explain how to derive the SDR problem (1) from the original OSTM problem. Starting from  OSTM:
>
> $$
> \begin{aligned}
> \max \quad & \sum_{(i, j) \in \Omega} \operatorname{tr}\left(O_{i}^{\top} C_{ij} O_{j}\right) \\\\
> \text{s.t.} \quad & O_{i} \in O_{d_i, r},  \quad i \in [n].
> \end{aligned}
> $$
>
> Let $X=[O_1; \cdots; O_n]$ and $U=XX^\top$. The objective function can be rewritten as
>
> $$
> \sum_{(i,j)\in\Omega} \operatorname{tr}(O_i^{\top} C_{ij} O_j) = \operatorname{tr}(X^{\top} C X) = \operatorname{tr}(C X X^{\top})=\langle C, U\rangle.
> $$
>
> Next, we derive  constraints for the new variable $U$.
> - $U$ must be positive semidefinite. This follows directly from the definition $U=XX^\top$.
> - $tr(U_{ii})=r$. This is because $tr(U_{ii})=tr(O_iO_i^\top)=tr(O_i^\top O_i)=r$.
> - $U_{ii} \preceq I_{d_i}$. This is because  $U_{ii} = O_iO_i^\top$ and the largest singular value of $ O_i$ is 1.
>
> Combining the new objective function and new constraints, we have
>
> $$
> \begin{aligned}
> \max_{U \succeq 0} \quad & \langle C,U\rangle \\\\
> \text{s.t.} \quad
> & \operatorname{tr}(U_{ii}) = r, \quad i = 1, \ldots, n, \\\\
> & U_{ii} \preceq I_{d_i}, \quad i = 1, \ldots, n.
> \end{aligned}
> $$
>
> ##  Motivation of data generation model (2)
> Model (2) is natural for generalized CCA in Won et al. (2022)[1]. Given several  high-dimensional random vectors, CCA tries to find their common latent random vector in a low-dimensional space that best explains their  common characteristics.
>
> Consider $n$ high-dimentional random vectors $a_i \in R^{d_i},i \in [n]$ driven by a common latent random vector $z\in\mathbb{R}^r$:
> $$a_i = G_i^* z + \epsilon_i$$
> where $G_i^\star \in O_{d_i,r}$ (non-random) form an orthogonal basis for an $r$-dimensional subspace of $R^{d_i}$. $\epsilon_i$ has mean zero and is uncorrelated with $z$ and $\epsilon_j$ for $j \ne i$. We further assume that the covariance  of $\epsilon_i$ is $\tau I_{d_i}$ and that $z$ has zero mean and covariance  $I_r$.  The goal of CCA is to infer $G^{\star}=\\{G_i^\star\\}\_{i \in [n]}$  by observing  $\\{a_i\\}\_{i \in [n]}$. To this end, for each $a_i$, we draw an i.i.d sample  denoted by $a_i^k=G_i^\star z^k + \epsilon_i^k, k \in [m]$ and define $A_i=[a_i^1,\cdots, a_i^m]$.    Since
>
> $$
> E[\frac{1}{m}A_iA_j^\top]= \begin{cases}
> G_i^\star G_j^{\star\top} & i \ne j, \\\\
> G_i^\star G_i^{\star\top}+\tau I_{d_i} & i=j,
> \end{cases}
> $$
>
> we define $W_{ij}=\frac{1}{m}A_iA_j^\top-E[\frac{1} {m}A_iA_j^\top]$. Then the MAXBET problem:
>
> $$
> \begin{aligned}
> \max \quad & \sum_{i, j} \operatorname{tr}\left(O_{i}^{\top}A_iA_j^\top O_j\right) \\\\
> \text{s.t.} \quad & O_{i} \in O_{d_i, r} , \quad i \in [n],
> \end{aligned}
> $$
>
> is equivalent to an OSTM problem with $C_{ij}=G_i^\star G_j^{\star\top}+W_{ij}$, which will recover $G^\star$ up to some global rotation. We will add this discussion to our revision.
>
> Besides, the $C=G^\star G^{\star \top}+\sigma W$ structure also arises in other applications such as cryo-EM[2] and multiview structure from motion[3].
>
> ## Importance of bound on $\sigma$
> The significance of our bound on $\sigma$ is twofold.
> - Our $O(\sqrt{n/\log n})$ bound for $\sigma$ fills the gap between empirical observation and theoretical prediction. Experimental results show the tightness of SDP relaxation and linear convergence of GPM under an $O(\sqrt{n})$ noise condition. However, theoretical analysis in previous literature only proves such results under an $O(n^{\frac{1}{4}})$ noise level, which is far more conservative than empirical results.
> - More importantly, our bound for $\sigma$ provides theoretical guarantees for using GPM in high-noise-level application scenarios. In real-world applications, dimension $n$ and noise level may both be very high. In this case, the previous $O(n^{\frac{1}{4}})$ noise level bound is insufficient to ensure convergence. However, our $O(\sqrt{n/\log n})$ bound provides significantly higher tolerance for noise level and offers theoretical guarantees for the usage of GPM.
>
>
>
> [1]: Orthogonaltrace-sum maximization:tightness of
> the semidefinite relaxation and guarantee of locally optimal solutions,Joong-ho Won et al.
>
> [2]: Viewing direction estimation in Cryo-EM using synchronization, Yoel Shkolnisky et al.
>
> [3]: Global motion estimation from point matches, M. Arie-Nachimson et al.

---

> > ### Author Rebuttal · Reviewer_nsWS · 2026-04-03
> >
> > I would like to thank the authors for their satisfactory explanations about the reviewers' comments. I raised my evaluation to accept.

---

### Decision · Program_Chairs · 2026-04-30

**Decision:**

Reject

**Comment:**

The paper studies a "multi-block" version of orthogonal trace sum maximization, maximizing \sum_{i,j} \Tr (O_i^T C_{ij} O_j) over orthogonal matrices O_i, O_j. For this the authors study both the SDP relaxation and a power method type algorithm and show improved convergence guarantees.

The reviewers all agree that the problem is worthy of study and the results are close to best possible. The main strength is that it's a clean improvement over prior results for a well-studied problem. Some of the reviews point out a few weaknesses: the writing, especially the motivation and comparison with prior results, can use some more work. So also, the novelty in techniques (specifically relative to the work of Bandeira et al. and other recent works that use the leave-one-out method) needs to be highlighted more. Overall, the paper is borderline.